# From Circuits to Dynamics: Understanding and Stabilizing Failure in 3D Diffusion Transformers

## Abstract

Reliable surface completion from sparse point clouds underpins many applications spanning content creation and robotics. While 3D diffusion transformers attain state-of-the-art results on this task, we uncover that they exhibit a catastrophic mode of failure: arbitrarily small on-surface perturbations to the input point cloud can fracture the output into multiple disconnected pieces – a phenomenon we call *meltdown*. Using activation-patching from mechanistic interpretability, we localize meltdown to a single early denoising cross-attention activation. We find that the singular-value spectrum of this activation provides a scalar proxy: its spectral entropy rises when fragmentation occurs and returns to baseline when patched. Interpreted through diffusion dynamics, we show that this proxy tracks a symmetry-breaking bifurcation of the reverse process. Guided by this insight, we introduce `PowerRemap`, a drop-in, test-time control that stabilizes sparse point-cloud conditioning. On Google Scanned Objects, `PowerRemap` has a stabilization rate of 98.3% for the state-of-the-art diffusion transformer WALA. Overall, this work is a case study on how diffusion model behavior can be understood and guided based on mechanistic analysis, linking a circuit-level cross-attention mechanism to diffusion-dynamics accounts of trajectory bifurcations.

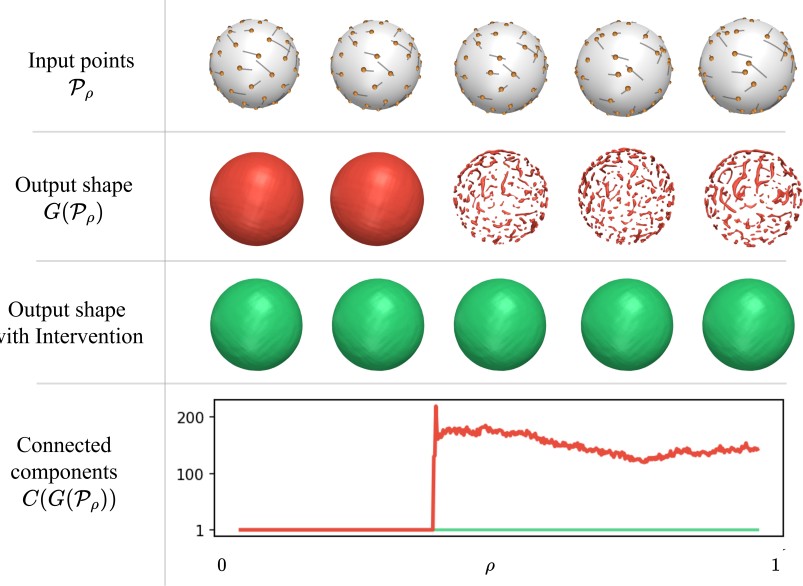

Figure 1: We investigate diffusion transformers on the task of surface reconstruction from sparse point clouds. We find that arbitrarily small on-surface perturbations to a point cloud can turn a shape into a speckle. We call this failure *meltdown* and study it through mechanistic interpretability and diffusion dynamics. Based on this analysis, we propose a test-time intervention, `PowerRemap`, which unlocks diffusion-based surface reconstruction under sparse conditions at test-time.

# 1 INTRODUCTION

From virtual content creation to dexterous manipulation, many core applications in vision and robotics hinge on reliable recovery of 3D surfaces from incomplete observations. This problem is called *surface reconstruction* from point clouds (Huang et al., 2022). In real-world applications, the available point clouds are often *sparse*, exacerbating the challenge of surface reconstruction. This sparsity motivates generative priors: *diffusion transformers* attain state-of-the-art results in generative tasks for many modalities (Chen et al., 2024; Sahoo et al., 2024; Jia et al., 2025; Lu et al., 2024b). Recently, they have been introduced to the 3D domain, overcoming challenges in surface reconstruction from sparse point clouds due to learned priors from large-scale datasets (Sanghi et al., 2024; Hui et al., 2024; Wu et al., 2024; Cao et al., 2024).

In this work, we investigate two state-of-the-art diffusion transformers for 3D surface reconstruction, WALA (Sanghi et al., 2024) and MAKE-A-SHAPE (Hui et al., 2024), on the task of surface reconstruction from sparse point clouds. We observe an intriguing *failure mode*: an imperceptible on-surface perturbation to the input point cloud can fracture the output into many disconnected pieces. We call this failure mode *meltdown* and analyze it through two lenses – *mechanistic interpretability* and *diffusion dynamics* – to understand its cause.

First, we employ *activation patching* to test the causal role of activations with respect to meltdown. We find that a single cross-attention activation, early in the denoising process, controls the failure. By investigating the effect of this activation, we find that its spectral entropy constitutes a *proxy* of the observed failure phenomenon. Second, we find that the identified proxy tracks a symmetry-breaking bifurcation of the reverse diffusion process. Based on these insights, we reverse-engineer a test-time intervention, `PowerRemap`, which unlocks diffusion-based surface reconstruction under sparse conditions at test-time.

Our contributions are summarized as follows:

1. **Failure Phenomenon: Meltdown**. We show that the state-of-the-art 3D diffusion transformers WALA (Sanghi et al., 2024) and MAKE-A-SHAPE (Hui et al., 2024) perform surface reconstruction from sparse point clouds in a brittle manner: small on-surface perturbations to the input point cloud can fracture the output into multiple disconnected pieces. We call this failure phenomenon *meltdown*.

2. **Interpretability**. This work provides a case study on how diffusion model behavior can be understood and guided based on mechanistic interpretability. We link a circuit-level cross-attention mechanism to diffusion-dynamics accounts of trajectory bifurcations.

3. **Test-Time Intervention: `PowerRemap`.** We propose a drop-in, test-time control to stabilize sparse point-cloud conditioning in diffusion transformers. `PowerRemap` averts meltdown in $98.3\%$ of cases on the *Google Scanned Object (GSO)* dataset (Downs et al., 2022) for WALA.

We introduce the failure phenomenon, meltdown, in Section 2. In Section 3, we analyze meltdown from the perspectives of mechanistic interpretability. Section 3.4 introduces our method, `PowerRemap` and presents results for it on the GSO dataset. Finally, we link meltdown to diffusion dynamics in Section 4 and discuss on current limitations in Section 6.

# 2 FAILURE PHENOMENON: MELTDOWN

We study a state-of-the-art diffusion transformer for 3D shape generation, namely WALA (Sanghi et al., 2024). In Appendix B, we show that many observations and insights transfer to another 3D diffusion transformer, namely MAKE-A-SHAPE (Hui et al., 2024). Such models can generate surfaces from several input modalities, including point clouds, thus solving the *surface reconstruction* task: given a set $\mathcal{P} = \{p_i\}_{i=1}^N \subset \mathcal{S} \subset \mathbb{R}^3$ of $N$ points sampled from an underlying surface $\mathcal{S}$, the model $G$ should reconstruct a surface consistent with the input and approximating the underlying surface $G(\mathcal{P}) \approx \mathcal{S}$. In many real-world scenarios (e.g., fast scene capture), $N$ can be small, i.e. the point cloud is *sparse*.

As illustrated in Figure 1, we observe that there exist two sparse point clouds $\mathcal{P}, \mathcal{Q}$ that are close in the input space, but the corresponding outputs differ severely: $G(\mathcal{P})$ is a connected surface while

$G(\mathcal{Q})$ is a fragmented "speckle" of disconnected pieces. We will refer to this sudden catastrophic fracture as *meltdown*.

To study this failure phenomenon systematically, let us first introduce the topological quantity $C$ that counts the connected components of the output surface and serves as a quantifiable identifier of the healthy ($C = 1$) versus unhealthy output ($C > 1$). Furthermore, let us consider a running example where the points are sampled from a simple sphere: $\mathcal{S} = \{x : \|x\|_2 = 1\}$. This allows us to perform experiments that precisely control for the distribution of the points. Specifically, we fix the random seed and first identify two point clouds of the same size $N = 400$: $\mathcal{P}_0$ which produces a sphere output $C(G(\mathcal{P}_0)) = 1$ and $\mathcal{P}_1$ which produces a speckle output $C(G(\mathcal{P}_1)) \gg 1$ (typically around 100). Using spherical interpolation (geodesics on general surfaces), we can construct a continuous family of point clouds $\mathcal{P}_\rho \subset \mathcal{S}$. We sweep $\rho \in [0, 1]$ and record $C(\rho) := C(G(\mathcal{P}_\rho))$.

Figure 1 illustrates the outcome of this experiment. As we sweep $\rho$ from 0 to 1, we first observe a long plateau of $C(\rho) = 1$, followed by a sudden jump to $C(\rho) \gg 1$ over a very narrow range of $\rho$. Refining the steps around this transition, we observe an effectively discontinuous jump in the macroscopic descriptor $C(\rho)$.

In Appendix B, we report observing meltdown across different diffusion transformers, i.e., WALA (Sanghi et al., 2024) and MAKE-A-SHAPE (Hui et al., 2024), denoising strategies, i.e., DDIM (Song et al., 2021) and DDPM (Ho et al., 2020b), and Google Scanned Objects (Downs et al., 2022).

## 3 MECHANISTIC INTERPRETATION AND INTERVENTION

After observing and quantifying *meltdown* in WALA, we ask: What is the root cause of this phenomenon? To address this question, we turn to the growing field of *mechanistic interpretability*.

Mechanistic interpretability (Geiger et al., 2021; Wang et al., 2023b; Sharkey et al., 2025) studies the internal mechanisms by which networks generalize, aiming to reverse-engineer representations so that model behavior can be predicted and guided. Core workflows of mechanistic interpretability involve decomposing models into analyzable components, and describing their roles and the flow of information between them. In this regard, *activation patching* (Heimersheim & Nanda, 2024; Zhang & Nanda, 2024) is one of the most prominent techniques. It tests the causal role of activations by swapping them between a *healthy* and a *unhealthy* run and measuring the respective outcome. In our context, we transition between a healthy (sphere) and unhealthy (speckle) run by continuously moving along a meltdown path quantified by our control parameter $\rho$. This enables us to transition between a healthy and unhealthy run in a controlled manner, which are ideal conditions for systematic activation patching to identify the root cause of meltdown.

### 3.1 WALA: DIFFUSION TRANSFORMER

Before we investigate the mechanistic behavior, we briefly summarize the relevant parts of the WALA diffusion transformer. A more detailed description is available in Appendix A, the original work (Sanghi et al., 2024), and the references therein.

**Transformer.** WALA is a latent diffusion model with a point-net encoder $E$, U-ViT-style (Hoogeboom et al., 2023) denoising backbone $B$, and VQ-VAE decoder (van den Oord et al., 2017) $D$. The U-ViT $B = B^{K-1} \circ \cdots \circ B^0$ has $K = 32$ transformer *blocks* $B^k$. The condition $\mathbf{C} \in \mathbb{R}^{1024 \times 1024}$ enters via both AdaLN modulation Esser et al. (2024) and cross-attention. Denoting by $\mathbf{Z}^k \in \mathbb{R}^{1728 \times 1152}$ the tokens entering the $k$-th block, it computes $B^k : \mathbf{Z}^k \mapsto \mathbf{Z}^{k+1}$ as a combination of multi-head self-attention SA and cross-attention CA layers (col. 2) with residual connections (col. 3):

$$\mathring{\mathbf{Z}} = \mathring{\text{AdaLN}}(\mathbf{Z}^k, \mathbf{C}), \qquad \mathring{\mathbf{Y}} = \text{SA}(\mathring{\mathbf{Z}}), \qquad \mathring{\mathbf{R}} = \mathring{\mathbf{Y}} + \mathbf{Z}^k, \qquad (1a)$$

$$\overset{\times}{\mathbf{Z}} = \overset{\times}{\text{AdaLN}}(\mathring{\mathbf{R}}, \mathbf{C}), \qquad \overset{\times}{\mathbf{Y}} = \text{CA}(\overset{\times}{\mathbf{Z}}, \mathbf{C}), \qquad \overset{\times}{\mathbf{R}} = \overset{\times}{\mathbf{Y}} + \mathring{\mathbf{R}}, \qquad (1b)$$

$$\bar{\mathbf{Z}} = \bar{\text{AdaLN}}(\overset{\times}{\mathbf{R}}, \mathbf{C}), \qquad \bar{\mathbf{Y}} = \text{MLP}(\bar{\mathbf{Z}}), \qquad \mathbf{Z}^{k+1} = \bar{\mathbf{Y}} + \overset{\times}{\mathbf{R}}. \qquad (1c)$$

**Diffusion.** WALA is trained in the standard DDPM (Ho et al., 2020b) framework. At inference, the reverse diffusion maps an initial Gaussian latent $\mathbf{Z}_T \sim \mathcal{N}(0, I)$ to $\mathbf{Z}_0$ by iterating over a fixed

**Algorithm 1** Localizing Meltdown via Activation Patching

---

**Require:** Encoder $E$; latent diffusion transformer $B$; decoder $D$; healthy point-cloud $\mathcal{P}$; unhealthy point-cloud $\mathcal{Q}$

1: $\mathbf{Z}_T^0 \sim \mathcal{N}(0, I)$                          ▷ sample initial noise

    **Record healthy activations:**

2: $\mathbf{C}_\mathcal{P} \leftarrow E(\mathcal{P})$                     ▷ embed the healthy point-cloud

3: **for** $t = T : 1$ **do**                         ▷ denoising loop

4:     **for** $k = 0 : K - 1$ **do**                   ▷ block loop

5:         $\mathbf{Z}_t^{k+1} \leftarrow B^k(\mathbf{Z}_t^k, \mathbf{C}_\mathcal{P})$ and record $\mathbf{Y}_{k,t}^{\text{healthy}} \leftarrow \overset{\times}{\mathbf{Y}}$

6:     **end for**

7:     $\mathbf{Z}_{t-1}^0 \leftarrow \text{DDIM}(\mathbf{Z}_t^{K-1})$             ▷ discrete denoising update

8: **end for**

    **Patch unhealthy activations:**

9: $\mathbf{C}_\mathcal{Q} \leftarrow E(\mathcal{Q})$                   ▷ embed the unhealthy point-cloud

10: **for** $t' = T : 1$ **do**                  ▷ denoising substitution loop

11:     **for** $k' = 0 : K - 1$ **do**           ▷ block substitution loop

12:         **for** $t = T : 1$ **do**             ▷ denoising loop

13:             **for** $k = 0 : K - 1$ **do**      ▷ block loop

14:                 $\mathbf{Z}_t^{k+1} \leftarrow B^k(\mathbf{Z}_t^k, \mathbf{C}_\mathcal{Q})$ but patch $\overset{\times}{\mathbf{Y}} \leftarrow \mathbf{Y}_{k,t}^{\text{healthy}}$ **if** $t' = t$ **and** $k' = k$

15:             **end for**

16:         $\mathbf{Z}_{t-1}^0 \leftarrow \text{DDIM}(\mathbf{Z}_t^{K-1})$        ▷ discrete denoising update

17:         **end for**

18:     $C_{k,t} \leftarrow C(D(\mathbf{Z}_0^{K-1}))$      ▷ decode shape and count connected components after patch

19:     **end for**

20: **end for**

21: **return** repair map $\{C_{k,t}\}_{k=0:K-1, t=1:T}$

---

schedule of denoising steps $t \in \mathcal{T} = \{T, \ldots, 0\}$, where at each step the denoiser conditioned on $\mathbf{C}$ updates $\mathbf{Z}_t \rightarrow \mathbf{Z}_{t-1}$. At inference-time, we can sample using DDIM (Song et al., 2021) or DDPM (Ho et al., 2020b).

## 3.2 LOCALIZING MELTDOWN VIA ACTIVATION PATCHING

In a diffusion transformer, activations span two axes: network depth (*blocks*) $k \in \mathcal{K} = \{0, \cdots, 31\}$ and diffusion time (*denoising steps*) $t \in \mathcal{T} = \{7, \cdots, 0\}$. This depth–time grid $\mathcal{K} \times \mathcal{T}$ constitutes the search space for the activation patching. In our search, we specifically target the token-wise cross-attention write $\overset{\times}{\mathbf{Y}} \in \mathbb{R}^{1728 \times 1152}$ which serves as the additive interface to the residual stream through which the context $\mathbf{C}$ influences the latent tokens. We scan the depth-time grid as given in Algorithm 1 and record healthy activations to patch them for forward passes based on an unhealthy point cloud.

We depict the result of the search procedure in Figure 2. We identify a *single* cross-attention write, $\mathbf{Y} \equiv \overset{\times}{\mathbf{Y}}_{4,7}$, to be responsible for meltdown. The location of this activation is consistent with prior work that finds cross-attention to have the strongest impact on the generated output in the *early* denoising time-steps (Liu et al., 2025). An intuition for this is that at early denoising steps the diffusion model overly relies on the conditioning as there is no other meaningful signal present.

The goal of mechanistic interpretability is to reverse-engineer internal mechanisms that are human-understandable functions. Since we observe the meltdown as we increase $\rho$, we ask whether there is an interpretable function of $\mathbf{Y}(\rho)$ that allows us to understand the mechanistic cause of meltdown.

## 3.3 INVESTIGATING THE EFFECT OF PATCHING

Having localized the failure to a single cross-attention write $\mathbf{Y}$, we ask which properties of this write predict meltdown and its rescue under patching. Empirically, we find a single scalar that captures the effect: the spectral entropy (Powell & Percival, 1979).

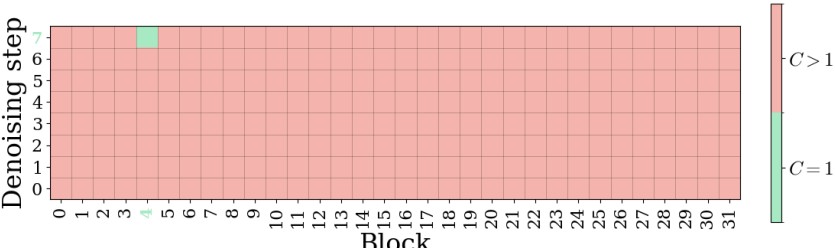

Figure 2: Our search in activation space finds that a single cross-attention write $\mathbf{Y}_{4,7}$ controls meltdown.

**Definition 1** (Spectral entropy). *Let $\sigma_i$ be the singular values of the matrix $\mathbf{Y}$. The spectral entropy of $\mathbf{Y}$ is*

$$H = -\sum_i p_i \log p_i \quad with \quad p_i = \frac{\sigma_i^2}{\sum_j \sigma_j^2} \tag{2}$$

*being the normalized directional energies.*

Low $H$ indicates that the update concentrates its energy in a few directions; high $H$ indicates a more isotropic, spread-out update. Figure 3a plots connectivity $C(\rho)$ and Figure 3b plots spectral entropy $H(\rho)$ along the same path $\rho$ for three diffusion seeds. In the baseline run, $H(\rho)$ increases roughly monotonically and smoothly with $\rho$; at some $\rho = \rho_{\mathrm{melt}}$ the surface fragments ($C > 1$). When we patch $\mathbf{Y}$ with its healthy value, $H(\rho)$ remains flat at the healthy level and the surface remains connected for all $\rho$. Thus $H(\rho)$ serves as a simple proxy that tracks failure (baseline) and rescue (patched) at the causal site.

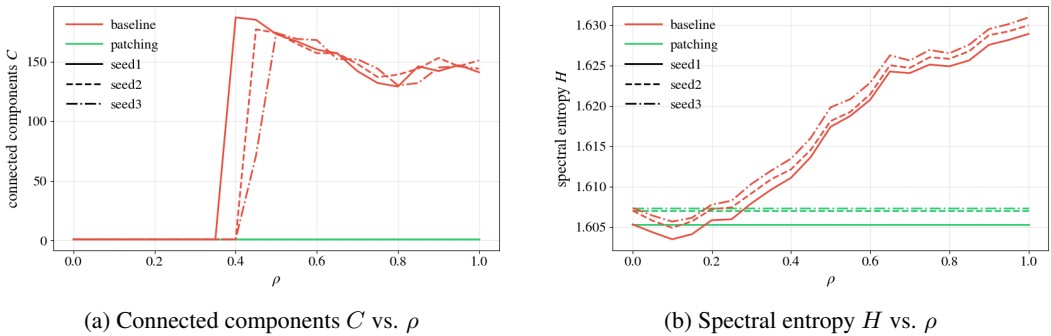

(a) Connected components $C$ vs. $\rho$           (b) Spectral entropy $H$ vs. $\rho$

Figure 3: As we move from a healthy to an unhealthy run, we observe that the baseline case shows a smooth rise in spectral entropy and a sudden jump in connectivity. Patching our $\mathbf{Y}$ keeps the spectral entropy at healthy levels and preserves connectivity. This behavior is consistent across diffusion seeds.

### 3.4 POWERREMAP: A TEST-TIME SPECTRAL INTERVENTION

Having identified the cross-attention write $\mathbf{Y}$ and the spectral entropy $H$ of its singular spectrum as a proxy that tracks failure and rescue, we now ask whether we can *directly* steer this quantity at test-time without access to healthy activations, which in-turn presumes access to a healthy point-cloud which is not the practice case. Our approach is simple: modify $\mathbf{Y}$ so that $H$ decreases while leaving the feature-directions (singular vectors of $\mathbf{Y}$) intact.

#### 3.4.1 METHOD

We introduce a minimal intervention that changes only the *magnitudes* of the singular values of $\mathbf{Y}$ and keeps its singular vectors fixed.

**Definition 2** (PowerRemap)**.** *Let* $\mathbf{Y} = \mathbf{U}\,\mathbf{\Sigma}\,\mathbf{V}^\top$ *be the SVD of* $\mathbf{Y}$ *with singular values* $\sigma_i \geq 0$. *Let* $\sigma_{\max} = \max_i \sigma_i > 0$ *and* $\gamma > 1$. *Then define*

$$\mathbf{\Sigma}' \;=\; \sigma_{\max}\,\left(\frac{\mathbf{\Sigma}}{\sigma_{\max}}\right)^\gamma, \qquad \mathit{PowerRemap}(\mathbf{Y}) \;=\; \mathbf{U}\,\mathbf{\Sigma}'\,\mathbf{V}^\top. \tag{3}$$

**Proposition 1** (PowerRemap *lowers spectral entropy*)**.** *Let* $H$ *and* $\mathit{PowerRemap}$ *be defined as above. For any* $\gamma > 1$,

$$H\big(\mathit{PowerRemap}(\mathbf{Y})\big) \;\leq\; H(\mathbf{Y}),$$

*with equality iff all* $\sigma_i > 0$ *are equal.*

This compresses the spectrum (smaller singular values shrink faster), which by construction targets the proxy parameter without changing the singular vectors of the features. We refer to the Appendix C for a corresponding proof.

### 3.4.2 EVALUATION AT SCALE

In this section, we assess whether the meltdown phenomenon and the effectiveness of PowerRemap generalize across diverse input geometries. To that end, we consider the *Google Scanned Objects (GSO)* (Downs et al., 2022) and *SimJEB* (Whalen et al., 2021) datasets and describe our evaluation protocol in Appendix B.3.

**GSO.** GSO (Downs et al., 2022) is a diverse corpus of 1,030 scanned household objects and was *not* used to train either WALA or MAKE-A-SHAPE. As depicted in Table 1 (top), we identify meltdown in 89.9% out of the 1,030 shapes and find that PowerRemap stabilizes failure in 98.3% of cases for the WALA model. We depict qualitative examples of the baseline meltdown and the corresponding PowerRemap in Figure 4. For MAKE-A-SHAPE, we identify meltdown in 88.3% out of the 1,030 shapes (cf. Table 4). We evaluate PowerRemap using a grid-search over $\gamma$ for a subset of 130 GSO shapes. We find that our method achieves a stabilization rate of 84.6% (cf. Table 2).

**SimJEB.** SimJEB (Whalen et al., 2021) is a curated benchmark of 381 3D jet-engine bracket CAD models that was *not* included in the training data of either WALA or MAKE-A-SHAPE. As depicted in Table 1, we identify meltdown in 92.4% out of the 381 shapes and find that PowerRemap stabalizes failure in 97.7% for the WALA model. For MAKE-A-SHAPE, we identify meltdown in 95.3% out of the 381 shapes (cf. Table 7). We evaluate PowerRemap using a grid-search over $\gamma$ over subset of 30 category-representative SimJEB shapes. We find that our method achieves a stabilization rate of 83.3% (cf. Table 2).

We note that we also experimented with alternative ways of reducing the spectral entropy, for example, reducing the temperature of the cross-attention block responsible for meltdown. However, those attempts did not alleviate the failure mode.

Table 1: Category-wise evaluation of `PowerRemap` on GSO and SimJEB (WALA) for a global $\gamma = 100$. Our method stabilizes failures in 98.3% of cases on GSO and 97.7% on SimJEB.

| Category | Shapes | Meltdown occurs [%] | `PowerRemap` rescues [%] | Avg. areal density |
|---|---|---|---|---|
| **GSO** | | | | |
| Shoe | 254 | 97.2 | 99.6 | – |
| Consumer goods | 248 | 97.6 | 99.2 | – |
| Unknown | 216 | 88.4 | 95.8 | – |
| Other | 112 | 92.9 | 99.0 | – |
| **Total** | 1030 | **89.9** | **98.3** | – |
| **SimJEB** | | | | |
| Arch | 37 | 89.2 | 100.0 | 1.225e-02 |
| Beam | 46 | 100.0 | 100.0 | 1.065e-02 |
| Block | 99 | 87.9 | 97.7 | 8.413e-03 |
| Butterfly | 43 | 93.0 | 95.0 | 1.282e-02 |
| Flat | 147 | 93.2 | 97.8 | 9.785e-03 |
| Other | 9 | 100.0 | 88.9 | 1.400e-02 |
| **Total** | 381 | **92.4** | **97.7** | 1.024e-02 |

Table 2: Category-wise evaluation of `PowerRemap` on a 130-shape subset of GSO and on a 30-shape subset of SimJEB (MAKE-A-SHAPE), using an adaptive-$\gamma$ strategy. Our method stabilizes failures in 84.6% of cases on the MAS subset and 83.3% on SimJEB (MAKE-A-SHAPE).

| Category | Shapes | Meltdown occurs [%] | `PowerRemap` rescues [%] | Avg. areal density |
|---|---|---|---|---|
| **GSO (subset)** | | | | |
| Consumer goods | 60 | 100.0 | 90.0 | – |
| Bottles, cans & cups | 23 | 100.0 | 95.7 | – |
| Unknown | 18 | 100.0 | 83.3 | – |
| Other | 29 | 100.0 | 65.5 | – |
| **Total** | 381 | **100.0** | **84.6** | – |
| **SimJEB (subset)** | | | | |
| Arch | 2 | 100.0 | 50.0 | 1.920e-02 |
| Beam | 4 | 100.0 | 75.0 | 1.856e-02 |
| Block | 9 | 100.0 | 100.0 | 1.371e-02 |
| Butterfly | 3 | 100.0 | 66.7 | 1.491e-02 |
| Flat | 11 | 100.0 | 90.9 | 1.370e-02 |
| Other | 1 | 100.0 | 0.0 | 8.706e-03 |
| **Total** | 30 | **100.0** | **83.3** | 1.488e-02 |

## 4 DIFFUSION DYNAMICS

*Diffusion dynamics* refers to a collection of ideas describing the generative diffusion process using established theory from statistical physics (Raya & Ambrogioni, 2023; Biroli et al., 2024; Yu & Huang, 2025; Ambrogioni, 2025), information theory (Ambrogioni, 2025), information geometry (Chen et al., 2023; Ventura et al., 2025), random-matrix theory (Ventura et al., 2025), and dynamical systems (Ambrogioni, 2025). Key concepts from diffusion dynamics allow us to frame both the observed failure phenomenon and the intervention, ultimately connecting the mechanistic analysis to a theoretically established interpretation of the generative diffusion process.

### 4.1 PRELIMINARIES

We introduce key ideas of diffusion dynamics adapted from Raya & Ambrogioni (2023); Biroli et al. (2024); Ambrogioni (2025). The reverse-time diffusion can be viewed as a noisy gradient flow in a time-dependent potential $u(\cdot, s)$:

$$d\mathbf{X}_t = -\nabla_{\mathbf{x}} u(\mathbf{X}_t, s)dt + g(s)d\mathbf{W}_t, \qquad u(\mathbf{x}, s) = -g^2(s)\log p(\mathbf{x}, s) + \Phi(\mathbf{x}, s), \qquad (4)$$

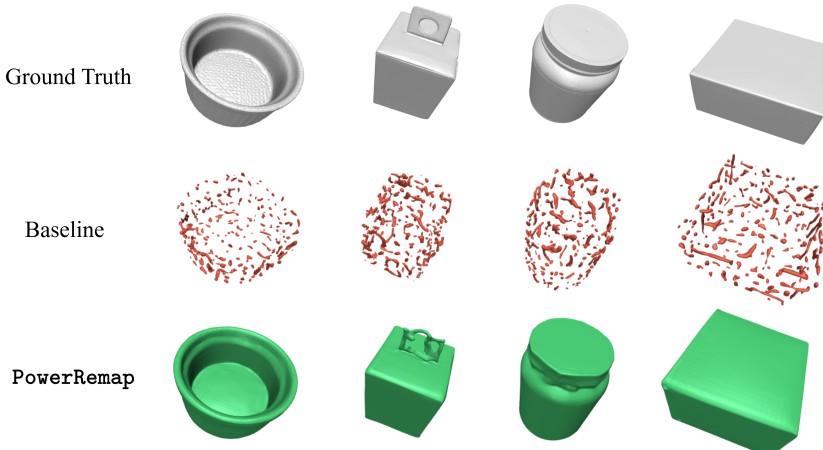

Ground Truth

Baseline

PowerRemap

Figure 4: Example results on the *Google Scanned Objects* dataset. We identify meltdown behavior in the WALA diffusion transformer for 89.9% of shapes. Out of these, the `PowerRemap` intervention rescues 98.3%, producing semantically valid outputs.

where $p(\cdot, s)$ is the forward marginal, $g$ is the noise scale, and $\Phi(\mathbf{x}, s) = \int_0^{\mathbf{x}} f(\mathbf{z}) d\mathbf{z}$ integrates the forward drift $f$. The potential $u$ is essentially a scaled and shifted marginal. The critical points $x^*$ of this potential $\nabla u(x^*, s) = 0$ are the *attractors* of the dynamics. Early in the generation ($t \approx 0$), there is a global symmetric basin with a stable central fixed point, and the trajectories exhibit mean-reverting fluctuations around it. As noise decreases, the energy landscape deforms and, at a critical time $\tau^*$, the fixed point loses stability and the landscape *bifurcates* into two basins. Such bifurcations repeat until at $t \approx T$ the potential has many fixed points aligning with the data modes (i.e., the data points under an exact score assumption). These bifurcation times $\tau^*$ can be interpreted as *decision* times where the sample trajectory is committed to a future attractor basin.

Around the degenerate critical point $x^*(\tau^*)$, two paths that are nearby for $t < \tau^*$ may diverge exponentially for $t > \tau*$ due to the Lyapunov exponent becoming positive (the smallest eigenvalue of $\nabla^2 u$ obtained from linearizing the reverse dynamics around the critical point). This can amplify tiny input differences and is the mechanism behind sending trajectories to different attractors.

This selection of one among many symmetry-equivalent states is called *spontaneous symmetry breaking*. A canonical example is a ferromagnet: at high temperature ($t \approx T$) spins are disordered, while as $t \to 0$ they align. Any magnetization direction is a priori equivalent, yet each realization picks one. The underlying symmetry is visible only in the ensemble over many realizations.

### 4.2 APPLICATION TO MELTDOWN AND INTERVENTION

To test the diffusion dynamic perspective of the meltdown phenomenon and the intervention, we perform several experiments predicted by this view. However, we must first introduce conditioning in the above diffusion dynamics view. For a fixed condition $\mathbf{C}$, this extension is trivial: simply modify the marginal $p(\cdot, s) = p(\cdot, s|\mathbf{C})$. However, a family of conditions, like the univariate interpolation $\{\mathbf{C}(\rho)|\rho \in [0, 1]\}$, introduces an additional dependence in the above formalism, and it is not obvious how to analyze the evident bifurcation around $\mathbf{C}^*$ instead of $\tau^*$. Fortunately, since the symmetry breaking originates *locally* around the bifurcation time $\tau^*$ and point $x^*$, a small change in the condition $d\mathbf{C}$ can be related to a small change in the initial condition $dx_T$ through the total differential of the reverse path $\gamma : (t, x_T, \mathbf{C}) \mapsto x_t$. Qualitatively, this allows us to consider different $x_T$ for a fixed $\mathbf{C}$ in place of different $\mathbf{C}$ for a fixed $x_T$.

**Ensemble.**   Spontaneous symmetry breaking suggests that even though a single trajectory commits to a single attractor (sphere versus speckle), both "symmetric" configurations are visited over an ensemble of random trajectories. We record the trajectories for 100 initial conditions $x_T \sim \mathcal{N}(0, I)$ over the $\rho \in [0, 1]$ range and plot the resulting shape connected component distribution in Figure 5a. The extremes $\rho = 0, 1$ are far from a critical condition, and all trajectories converge to the

respective attractors. However, at the intermediate conditions, the ensemble of trajectories visits both attractors, with the ratio of fractured shapes increasing steadily with $\rho$. In expectation, the component curve $\mathbb{E}_{x_T}[C(\rho)]$ exhibits a smooth behavior, relaxing the discrete jump in $C(\rho)$ for a single $x_T$.

**Trajectories.** We can illustrate and qualitatively confirm the explanation for meltdown suggested by the diffusion dynamics perspective by considering an intermediate $\rho = 0.4$ for which the ensemble visits both attractors. This is visualized in Figure 5b for 1000 trajectories, projected onto a 2D linear subspace spanned by the first two principal components of the final distribution $p(\cdot, 0)$, which shows two major modes and some minor modes. The baseline trajectories are colored blue/red for sphere/speckle results. The initial distribution $p(\cdot, T)$ is a Gaussian from which all the trajectories originate. The first denoising step resembles the initial mean-reverting stage of denoising (Biroli et al., 2024; Ventura et al., 2025) as the trajectories remain close. The second step marks the symmetry breaking. We can also imagine a "decision boundary" between the blue and red trajectories: this is the projected *separatrix* that demarcates the two attractor basins. The intervention alters the first step, after which the close bundle of trajectories flows smoothly to a tight minor mode of the baseline.

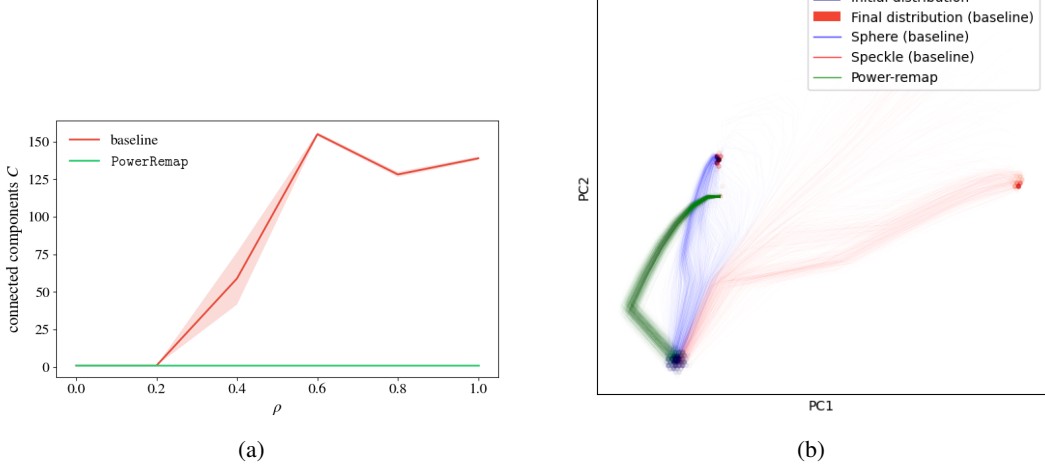

(a)                                (b)

Figure 5: A collection of diffusion trajectories reveals additional insights about the meltdown phenomenon. (a) In expectation over the initial noise, both the sphere and speckle shapes are produced at intermediate conditions, relaxing the sharp meltdown behavior for a fixed initial noise. (b) Latent diffusion trajectories projected onto a 2D linear subspace spanned by the first two principal components of the final distribution of the baseline. The `PowerRemap` trajectories in green form a tight bundle following a different path that converges to a minor mode of the baseline distribution.

**Potential.** We calculate the potential similar to the procedure introduced by Raya & Ambrogioni (2023). We select a pair of representative trajectories from each attractor and interpolate between them along a variance-preserving curve $x_t(\alpha) = \cos(\alpha)x_t^{\text{sphere}} + \sin(\alpha)x_t^{\text{speckle}}$ for $\alpha \in [-0.2\pi, 1.2\pi]$. Figure 6 reveals the two diffusion stages separated by the bifurcation time $\tau^* \approx 5$, where the single potential well flattens and splits into the two attractor basins.

## 5 RELATED WORK

**Activation patching** In the literature, activation patching has been used to locate mechanistic modules of causal interest in many modalities, ranging from language (Wang et al., 2023a; Meng et al., 2022; Conmy et al., 2023) to vision-language (Golovanevsky et al., 2025) and audio (Facchiano et al., 2025). To the best of our knowledge, this is the first work that leverages activation patching in the geometry domain. The differentiating factor of geometry over the modalities mentioned before is that it is objectively measurable at the output (connectedness) as well as at the input

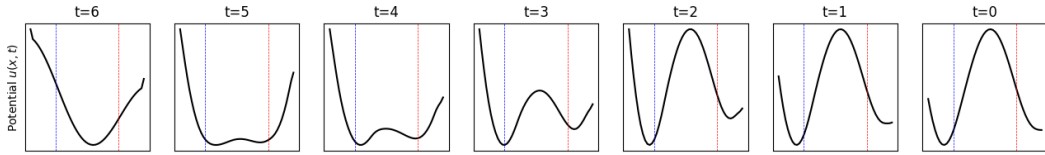

Figure 6: The potential $u$ (related to the marginal probability via Eq. (4)) reveals the two diffusion stages separated by the bifurcation time $\tau^* \approx 5$, where the single potential well flattens and splits into the two attractor basins. The particle's location just before this early bifurcation commits it to the final attractor and ultimately determines the generated shape. Small perturbations around this time become amplified, giving the appearance of discrete jumps that characterize the observed meltdown.

(point cloud). Using point-cloud inputs as opposed to text prompts allows us to continuously interpolate between a healthy and an unhealthy run. As a result, we can quantify how the causal influence of an identified site changes smoothly along this interpolation.

**Spectral entropy.** Spectral measures are increasingly used in interpretability. Skean et al. (2025) study the matrix entropy of hidden layers, while Yunis et al. (2024) and Lu et al. (2024a) relate weight spectra to generalization. Our analysis provides further evidence that the spectral properties of model internals are indicative of behavior.

## 6 DISCUSSION AND LIMITATIONS

While we identify a simple interpretable proxy for the meltdown (spectral entropy) and also reduce the effect of activation patching to a simple intervention (`PowerRemap`), we cannot argue for the uniqueness of these, for a concrete example, how to select the strength $\gamma$ of the remap. There exists a phenomenological hierarchy: the generated shape, the diffusion process, and the transformer circuits. While we link the sharp meltdown behavior to bifurcations in the diffusion dynamics, and a specific circuit to the generated shape, a question remains: *why* does a decreased spectral entropy reduce invalid outputs? From the SVD of the **Y** matrix, which stacks outputs of all cross-attention heads, the first singular vector is by construction the direction that maximizes the shared variance across heads and can thus be interpreted as a feature identified by multiple heads. Correspondingly, the relative boost of this dominant shared direction may facilitate a "consensus" which is required for a valid output. A fully satisfactory answer would presume this link to be universal across models, not a mere particularity of the circuit. Preliminary results on the MAKE-A-SHAPE model suggest the link between the spectral entropy and the validity of the output transfers to a large degree, but more work is needed to draw strong conclusions. Overall, we find that the optimal `PowerRemap` strength $\gamma$ is model-dependent.

## 7 CONCLUSION

We identified an intriguing failure mode in the state-of-the-art diffusion transformer WALA (Sanghi et al., 2024) for 3D surface reconstruction, namely meltdown: small on-surface perturbations to the input point cloud result in fragmented output shapes. This failure mode can be traced back to activations after a cross-attention branch at early stages of the reverse diffusion process. We found that the spectral entropy of the activations is an indicator of impending meltdown. Based on these insights, we proposed a simple but efficient test-time remedy, `PowerRemap`, that reduces spectral entropy and successfully rescues meltdown in the majority of cases on the widely used *Google Scanned Objects* (Downs et al., 2022) dataset. While our analysis was derived from Sanghi et al. (2024), we observed that meltdown and its mechanistic cause transfer to another 3D diffusion transformer, MAKE-A-SHAPE (Hui et al., 2024), suggesting broader relevance beyond a single architecture. In addition to this practical remedy, we established a connection between meltdown and bifurcations in diffusion dynamics, offering a mechanistic lens on how instabilities arise during the reverse process. We believe this perspective not only advances the robustness of 3D surface reconstruction but also opens new avenues for interpretability research in diffusion models.

## REPRODUCIBILITY STATEMENT

We describe all experimental setups in the main text. Appendix B provides exact reproduction protocols and lists the random seeds for every result.

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

# A  BACKGROUND

In this section, we provide background information on the diffusion transformers WALA (Sanghi et al., 2024) and MAKE-A-SHAPE (Hui et al., 2024) along the dimensions *diffusion* (Appendix A.1) and *transformer* (Appendix A.2).

## A.1  DIFFUSION

**Forward transition.** Diffusion generative models synthesize data by inverting a *Markov* chain that gradually corrupts an observation $\mathbf{x}_0 \sim p_{\text{data}}$ with Gaussian noise over $T$ discrete timesteps (Sohl-Dickstein et al., 2015; Ho et al., 2020a). The forward (noising) transition is

$$q(\mathbf{x}_t \mid \mathbf{x}_{t-1}) = \mathcal{N}\big(\mathbf{x}_t; \sqrt{1 - \beta_t}\,\mathbf{x}_{t-1},\ \beta_t \mathbf{I}\big), \quad t = 1, \ldots, T, \tag{5}$$

$$q(\mathbf{x}_t \mid \mathbf{x}_0) = \mathcal{N}\big(\mathbf{x}_t; \sqrt{\bar{\alpha}_t}\,\mathbf{x}_0,\ (1 - \bar{\alpha}_t)\mathbf{I}\big), \quad \text{with} \quad \bar{\alpha}_t = \prod_{s=1}^{t}(1 - \beta_s), \tag{6}$$

where the variance schedule $\{\beta_t\}_{t=1}^{T} \subset (0,1)$ is chosen so that $\mathbf{x}_T$ is nearly *i.i.d.* $\mathcal{N}(0, I)$. Both architectures are associated with a cosine variance schedule.

**Noise prediction objective.** Instead of directly regressing $\mathbf{x}_0$, the denoising neural networks $\epsilon_\theta$ of WALA and MAKE-A-SHAPE have been trained to predict the added noise:

$$\mathcal{L}_{\text{simple}}(\theta) = \mathbb{E}_{t,\mathbf{x}_0,\epsilon}\Big[\big\|\epsilon - \epsilon_\theta(\underbrace{\sqrt{\bar{\alpha}_t}\,\mathbf{x}_0 + \sqrt{1 - \bar{\alpha}_t}\,\epsilon}_{\mathbf{x}_t},\ t)\big\|_2^2\Big], \qquad \epsilon \sim \mathcal{N}(0, I). \tag{7}$$

This "$\epsilon$-parameterization" empirically stabilizes training and is adopted by nearly all modern models (Ho et al., 2020a).

**Denoising (DDPM).** Given a trained $\epsilon_\theta$, the original *Denoising Diffusion Probabilistic Model* (DDPM) (Ho et al., 2020a) samples via the stochastic reverse transition

$$p_\theta(\mathbf{x}_{t-1} \mid \mathbf{x}_t) = \mathcal{N}\big(\mathbf{x}_{t-1}; \underbrace{\frac{1}{\sqrt{1 - \beta_t}}\Big(\mathbf{x}_t - \frac{\beta_t}{\sqrt{1 - \bar{\alpha}_t}}\,\epsilon_\theta(\mathbf{x}_t, t)\Big)}_{\boldsymbol{\mu}_\theta(\mathbf{x}_t, t)},\ \sigma_t^2 \mathbf{I}\big), \qquad \sigma_t^2 = \tilde{\beta}_t, \tag{8}$$

where $\tilde{\beta}_t = \beta_t \frac{1 - \bar{\alpha}_{t-1}}{1 - \bar{\alpha}_t}$. Iterating Eq. equation 8 from $t = T$ to 1 produces $\mathbf{x}_0$ in $T$ noisy steps.

**Denoising (DDIM).** Song et al. (2021) showed that the same model admits a *deterministic* implicit sampler (DDIM) obtained by setting the variance term to zero:

$$\mathbf{x}_{t-1} = \sqrt{\bar{\alpha}_{t-1}}\,\underbrace{\left(\frac{\mathbf{x}_t - \sqrt{1 - \bar{\alpha}_t}\,\epsilon_\theta(\mathbf{x}_t, t)}{\sqrt{\bar{\alpha}_t}}\right)}_{\hat{\mathbf{x}}_0} + \sqrt{1 - \bar{\alpha}_{t-1}}\,\epsilon_\theta(\mathbf{x}_t, t). \tag{9}$$

Eq. equation 9 preserves the marginal $q(\mathbf{x}_{t-1} \mid \mathbf{x}_0)$, enabling user-specified *inference* schedules (e.g., $t = T, \ldots, 1$ with $T \gg 1$ for high fidelity or sparse subsets for speed) without retraining. Crucially, Eqs. equation 8–equation 9 share the same $\epsilon_\theta$ trained via Eq. equation 7. Hence one can *train* with the log-likelihood–consistent DDPM objective but *sample* using DDPM or DDIM.

**Classifier-free guidance.** Both architectures employ *classifier-free guidance* (CFG) (Ho & Salimans, 2022). CFG biases the denoising direction toward a user condition without requiring an external classifier. For a current latent $\mathbf{x}_t$ and the shared noise predictor $\epsilon_\theta$, we obtain two estimates at the same step $t$: the unconditional prediction $\epsilon_{\text{uncond}}$ (with the condition omitted) and the conditional prediction $\epsilon_{\text{cond}}$ (under the desired condition). We then form a guided estimate

$$\tilde{\epsilon} = \epsilon_{\text{uncond}} + s\big(\epsilon_{\text{cond}} - \epsilon_{\text{uncond}}\big), \qquad s \geq 0,$$

and substitute $\tilde{\epsilon}$ in place of $\epsilon_\theta(\mathbf{x}_t, t)$ in the DDPM/DDIM updates (Eqs. equation 8–equation 9). Setting $s = 1$ at inference-time ignores updates from the unconditional stream, i.e., $\tilde{\epsilon} = \epsilon_{\text{cond}}$.

## A.2 TRANSFORMER

The noise–prediction objective equation 7 only specifies what to learn but leaves open ow the denoiser $\epsilon_\theta$ is parameterised. Classical DDPMs adopt a convolutional UNET encoder–decoder (Ronneberger et al., 2015; Ho et al., 2020a), whereas modern large-scale models (e.g. Stable Diffusion, Imagen) replace the convolutional blocks with *Transformer* layers, yielding the **UViT** ("U-shaped Vision Transformer") backbone that now dominates high-fidelity diffusion systems(Bao et al., 2023; Karras et al., 2022). Both architectures, WALA and MAKE-A-SHAPE, implement a *diffusion generative model* via *Transformer* layers.

### A.2.1 OVERVIEW

Both methods adopt a wavelet–latent diffusion pipeline in which 3D shapes are represented as multiscale wavelet coefficients and a U-ViT-style denoising backbone (Hoogeboom et al., 2023) is trained in the DDPM (Ho et al., 2020b) framework. The key difference lies in how the wavelet data are fed to the diffusion core.

1. WALA first compresses the full wavelet tree with a convolutional VQ-VAE (stage 1), mapping the diffusible wavelet tree to a latent grid. The latent grid is then modeled by a 32-layer U-ViT (stage 2), where each Transformer layer runs self-attention and cross-attention, totaling 32 cross-attention calls.

2. MAKE-A-SHAPE skips the auto-encoder and instead packs selected wavelet coefficients into a compact grid. The U-ViT backbone then downsamples this tensor to a bottleneck volume. The bottleneck is traversed by a 16-layer U-ViT core—8 self-attention layers immediately followed by 8 cross-attention layers— before up-sampling restores the packed grid

### A.2.2 CONDITIONING PATHWAY (POINT-CLOUD)

In general, both MAKE-A-SHAPE and WALA share a common pipeline: PointNet encoding followed by aggregation and injecting the resulting latent vectors into the U-ViT generator via *(i)* affine modulation of normalization layers and *(ii)* cross-attention.

In particular, MAKE-A-SHAPE injects the conditioning latent vectors into the U-ViT generator at three stages: (1) *concatenation*: the latent vectors are aggregated and concatenated as additional channels of the input noise coefficients, (2) *affine modulation*: the latent vectors are aggregated and subsequently utilized to condition the convolution (down-sampling) and deconvolution (up-sampling) layers via modulating the affine parameters of the group normalization layers, (3) *cross-attention*: each condition latent vector is augmented with an element-wise positional encoding and then fed into a cross-attention module alongside the bottleneck volume.

WALA injects the conditioning latent vectors into the U-ViT generator at two stages: (1) *affine modulation*: the latent vectors are linearly projected via a global projection network and used to modulate the scale and bias parameters of GroupNorm layers in both the ResNet and attention blocks (AdaGN) (Esser et al., 2024), (2) *cross-attention*: each latent vector, augmented with an element-wise positional encoding, is employed as the key and value in cross-attention modules interleaved within each transformer block.

## B EXPERIMENTS

This section shows that many observations and insights gained through studying the diffusion transformer WALA (Sanghi et al., 2024) under DDIM sampling on spheres transfer (i) to the diffusion transformer MAKE-A-SHAPE (Hui et al., 2024) (ii) sampling under DDPM and (iii) other shapes (GSO and SimJEB). Additionally, this section details the experimental setup to reproduce our results. In particular:

1. GENERAL (B.1): This section provides an overview on our experimental setup.
2. SPHERE EXPERIMENTS (B.2):
   (a) GENERAL (B.2.1: This section provides an overview on the setup for the sphere experiments.
   (b) WALA, DDIM (B.2.2): This section reports the experimental setup for the sphere experiments in the main text.
   (c) WALA, DDPM (B.2.3): This sections reports additional results for WALA under DDPM sampling.
   (d) MAKE-A-SHAPE, DDIM (B.2.4): This section provides results for MAKE-A-SHAPE under DDIM sampling.
   (e) MAKE-A-SHAPE, DDPM B.2.5: This section provides results for MAKE-A-SHAPE under DDPM sampling.
3. Datasets: GOOGLE SCANNED OBJECTS (GSO) (B.3) and SIMJEB: This section details our experiments on GSO (Downs et al., 2022) and SimJEB (Whalen et al., 2021).

### B.1 GENERAL

This section provides a general overview on the experimental setup for all results reported in this work.

**Restrict Analysis to Conditional Stream.** As the failure behavior, *meltdown*, is independent of the unconditional stream, we exclusively investigate the conditional prediction stream. That is, we set the CFG scale $s = 1.0$ and restrict our mechanistic analysis (e.g. activations) and diffusion dynamics analysis (e.g., latents $x_T$) to the conditional stream.

**Seeding.** Randomness regarding a diffusion trajectory is controlled globally by seeding Python, NumPy, and PyTorch (`torch.backends.cudnn.deterministic=True`, `benchmark=False`) so that every evaluation at a given $\rho$ starts from the same terminal noise $x_T$.

### B.2 SPHERE EXPERIMENTS

This section (i) provides a detailed account on our setup for the sphere experiments and (ii) reports additional results for *meltdown* on MAKE-A-SHAPE and DDPM sampling.

### B.2.1 SETUP

We detail the minimal, fully reproducible setup used to produce the sphere experiments for WALA and MAKE-A-SHAPE. Throughout, the control parameter is $\rho \in [0, 1]$, and the phenomenological order parameter is the number of connected components $C(\rho)$ in the generated mesh.

**Conditioning clouds on the sphere.** We work on $\mathcal{S} = \{x : \|x\|_2 = 1\}$ and fix $N = 400$ points for WALA and $N = 1200$ for MAKE-A-SHAPE. The base cloud $\mathcal{P}(0)$ uses a golden-angle (Fibonacci) sphere distribution:

$$g = \pi(3 - \sqrt{5}), \quad i \in \{0, \ldots, N - 1\}, \quad y_i = 1 - \frac{2(i + 0.5)}{N}, \quad r_i = \sqrt{1 - y_i^2}, \quad \theta_i = g\,i,$$

$$p_i(0) = \left(r_i \cos \theta_i,\ y_i,\ r_i \sin \theta_i\right).$$

A second target cloud $\mathcal{P}(1)$ is produced by jittering each $p_i(0)$ with i.i.d. Gaussian noise $n_i \sim \mathcal{N}(0, 0.1^2 \mathbf{I}_3)$ and renormalizing to the unit sphere:

$$\tilde{p}_i = \frac{p_i(0) + n_i}{\|p_i(0) + n_i\|_2}, \qquad \mathcal{P}(1) = \{\tilde{p}_i\}_{i=1}^N.$$

We then move each point *along* the surface via per-point spherical linear interpolation (SLERP) between corresponding pairs:

$$p_i(\rho) = \text{slerp}\big(p_i(0), \tilde{p}_i; \rho\big) = \frac{\sin((1 - \rho)\omega_i)}{\sin \omega_i} p_i(0) + \frac{\sin(\rho\,\omega_i)}{\sin \omega_i} \tilde{p}_i, \quad \omega_i = \arccos\big(\langle p_i(0), \tilde{p}_i \rangle\big).$$

This yields the cloud path $\mathcal{P}(\rho) = \{p_i(\rho)\}_{i=1}^N$ used throughout.

**Decoding and component counting.** Given $\mathcal{P}(\rho)$, we compute a conditioning code via the model's encoder and sample a latent with the diffusion sampler to yield $G(\mathcal{P}(\rho))$, i.e. a `mesh`. We report

$$C(\rho) = \text{len}(\texttt{trimesh.split(mesh)}),$$

i.e., the number of connected components in `trimesh`.

**Grid over the control parameter.** W sweep a uniform grid of $\rho$ values, i.e., $\rho \in \{0, 0.05, \dots, 1.0\}$.

**Connectivity curve $C(\rho)$.** We evaluate $C(\rho)$ on the uniform $\rho$ grid. For each $\rho$, we reseed the RNGs to reproduce the identical terminal noise $x_T$. The curve reported is is the set

$$\{(\rho, C(\rho))\}_{\rho \in \{0, 0.05, \dots, 1.0\}},$$

from which the observed plateau at $C(\rho) = 1$ and the subsequent jump to $C(\rho) > 1$ over a narrow $\rho$-interval (a connectivity bifurcation) are directly obtained.

**Spectral entropy curve $H(\rho)$.** We evaluate the spectral entropy of the localized cross–attention write on the same uniform control grid $\rho \in \{0, 0.05, \dots, 1.0\}$ and with identical terminal noise across $\rho$.

For each $\rho$, we encode the cloud $\mathcal{P}(\rho)$, run a single sampling trace, and read out the token-wise cross-attention write at the chosen site, $\mathbf{Y}(\rho)$. Let $\{\sigma_i(\rho)\}_i$ be the singular values of $\mathbf{Y}(\rho)$ (SVD of the matrix with shape tokens $\times$ features). We form normalized directional energies

$$p_i(\rho) = \frac{\sigma_i(\rho)^2}{\sum_j \sigma_j(\rho)^2}, \qquad H(\rho) = -\sum_i p_i(\rho) \log p_i(\rho),$$

using the natural logarithm. The reported curve is the set

$$\big\{ (\rho, H(\rho)) \big\}_{\rho \in \{0, 0.05, \dots, 1.0\}}.$$

### B.2.2 WALA, DDIM

**Key hyperparameters.**

| | |
|---|---|
| Model | `ADSKAILab/WaLa-PC-1B` |
| Sampler | DDIM ($\eta = 0$) |
| Diffusion rescale | 8 steps (`diffusion_rescale_timestep=8`), i.e., default |
| CFG weight | 1.0 (`scale=1.0`), i.e., we consider only conditional stream |
| Points per cloud | $N = 400$ |
| Cloud source | Unit sphere, golden-angle placement |
| Target cloud | Gaussian jitter $\sigma = 0.1$ on $\mathbb{R}^3$, renormalize to $\mathbb{S}^2$ |
| Interpolation | Per-point SLERP, control $\rho \in [0, 1]$ |
| $\rho$ grid | 21 values: $0, 0.05, \dots, 1.0$ |
| Seeds | $0 - 1000$ for all RNG calls |
| Order parameter | $C(\rho) = \#$ connected components (`trimesh.split`) |
| Device | `cuda` (CPU is functionally equivalent but slower) |

### B.2.3 WALA, DDPM

The activation-patching grid for WALA under DDPM is equivalent to Figure 2, i.e., WALA under DDIM. The corresponding for curves can be found in Figure 7.

**Key hyperparameters.**

| | |
|---|---|
| Model | `ADSKAILab/WaLa-PC-1B` |
| Sampler | DDPM |
| Diffusion rescale | 8 steps (`diffusion_rescale_timestep=8`) |
| CFG weight | 1.0 (`scale=1.0`) (we consider only conditional stream) |
| Points per cloud | $N = 400$ |
| Cloud source | Unit sphere, golden-angle placement |
| Target cloud | Gaussian jitter $\sigma = 0.1$ on $\mathbb{R}^3$, renormalize to $\mathbb{S}^2$ |
| Interpolation | Per-point SLERP, control $\rho \in [0, 1]$ |
| $\rho$ grid | 21 values: $0, 0.05, \ldots, 1.0$ |
| Seeds | 0 for all RNG calls |
| Order parameter | $C(\rho) = \#$ connected components (`trimesh.split`) |
| Device | `cuda` (CPU is functionally equivalent but slower) |

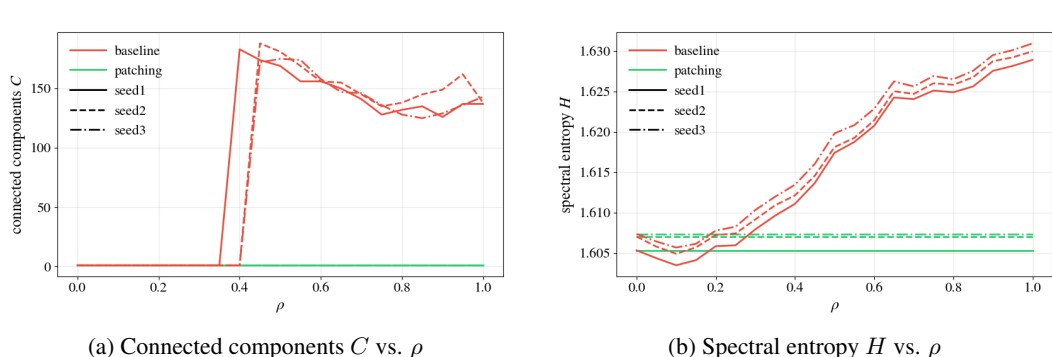

(a) Connected components $C$ vs. $\rho$         (b) Spectral entropy $H$ vs. $\rho$

Figure 7: [WALA, DDPM]. Our results from WALA under DDIM sampling transfer to WALA under DDPM sampling.

### B.2.4 MAKE-A-SHAPE, DDIM

We report the result for the activation search procedure for MAKE-A-SHAPE under DDIM in Figure 8. The corresponding curves are depicted in Figure 9.

**Key hyperparameters.**

| | |
|---|---|
| Model | `ADSKAILab/Make-A-Shape-point-cloud-20m` |
| Sampler | DDIM |
| Diffusion rescale | 100 steps (`diffusion_rescale_timestep=8`), i.e., default |
| CFG weight | 1.0 (`scale=1.0`), i.e., we consider only conditional stream |
| Points per cloud | $N = 1200$ |
| Cloud source | Unit sphere, golden-angle placement |
| Target cloud | Gaussian jitter $\sigma = 0.1$ on $\mathbb{R}^3$, renormalize to $\mathbb{S}^2$ |
| Interpolation | Per-point SLERP, control $\rho \in [0, 1]$ |
| $\rho$ grid | 21 values: $0, 0.05, \ldots, 1.0$ |
| Seeds | 0 for all RNG calls |
| Order parameter | $C(\rho) = \#$ connected components (`trimesh.split`) |
| Device | `cuda` (CPU is functionally equivalent but slower) |

### B.2.5 MAKE-A-SHAPE, DDPM

The activation-patching grid for MAKE-A-SHAPE under DDPM is equivalent to Figure 8, i.e., MAKE-A-SHAPE under DDIM. The corresponding curves can be found in Figure 10.

**Key hyperparameters.**

| | |
|---|---|
| Model | `ADSKAILab/Make-A-Shape-point-cloud-20m` |
| Sampler | DDPM |
| Diffusion rescale | 100 steps (`diffusion_rescale_timestep=8`), i.e., default |
| CFG weight | 1.0 (`scale=1.0`) (we consider only conditional stream) |
| Points per cloud | $N = 1200$ |
| Cloud source | Unit sphere, golden-angle placement |
| Target cloud | Gaussian jitter $\sigma = 0.1$ on $\mathbb{R}^3$, renormalize to $\mathbb{S}^2$ |
| Interpolation | Per-point SLERP, control $\rho \in [0, 1]$ |
| $\rho$ grid | 21 values: $0, 0.05, \dots, 1.0$ |
| Seeds | 0 for all RNG calls |
| Order parameter | $C(\rho) = \#$ connected components (`trimesh.split`) |
| Device | `cuda` (CPU is functionally equivalent but slower) |

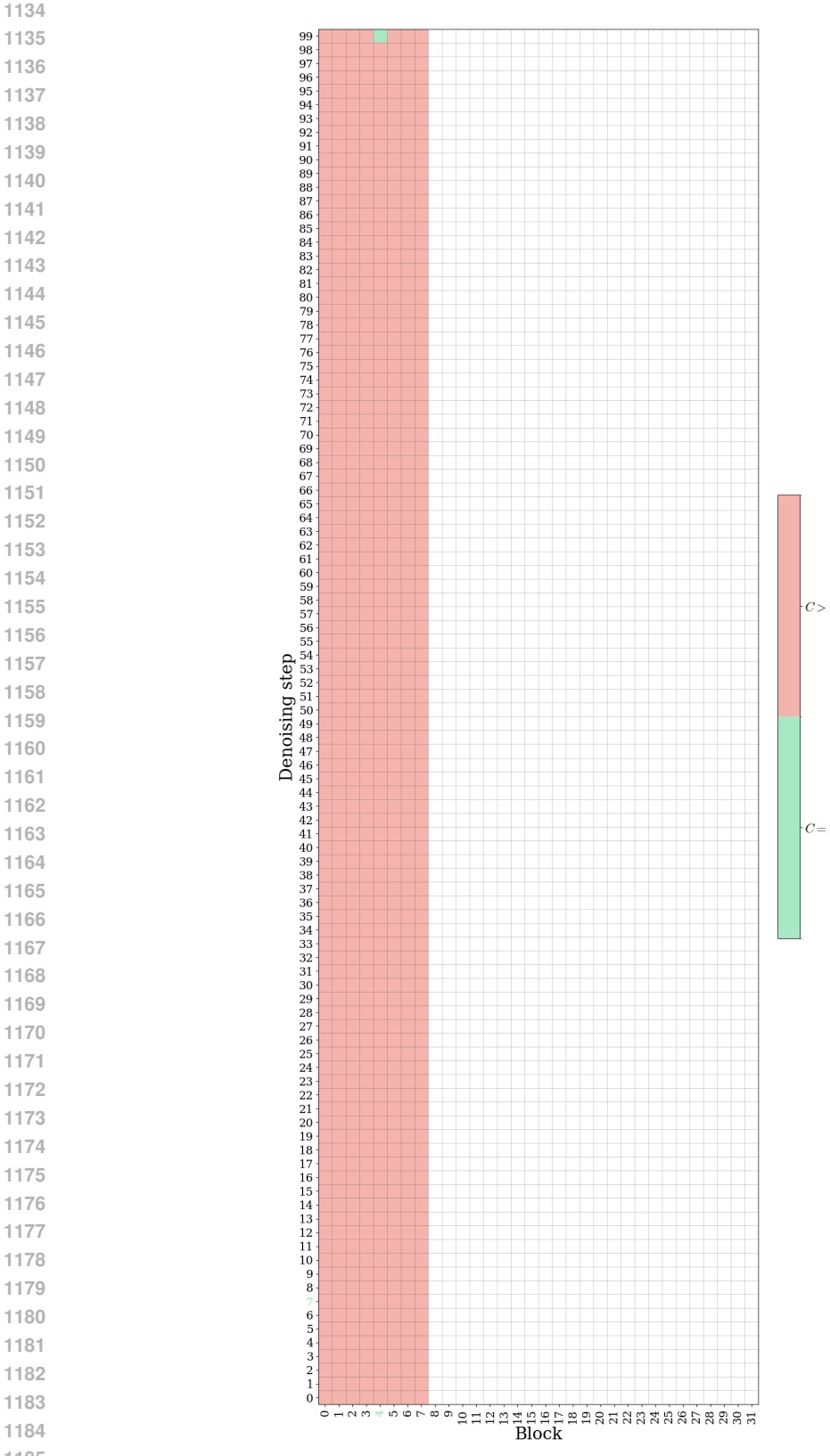

Figure 8: Activation-patching result for MAKE-A-SHAPE. Analogous to our result for WALA, we find an early denoising cross-attention activation that controls meltdown behavior.

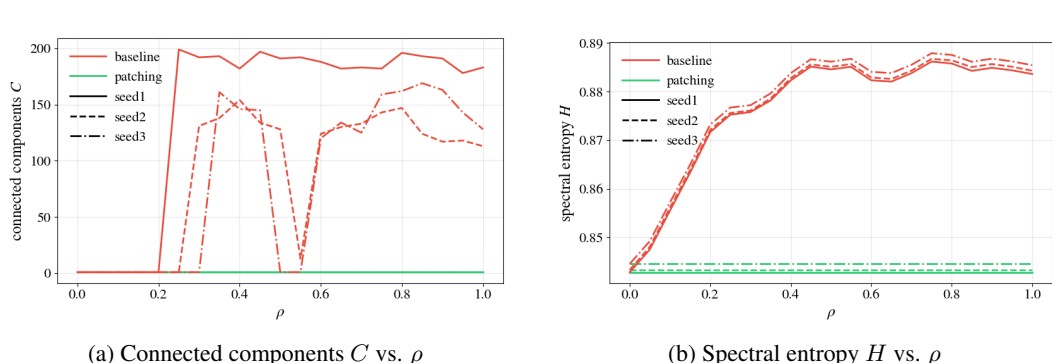

(a) Connected components $C$ vs. $\rho$    (b) Spectral entropy $H$ vs. $\rho$

Figure 9: [MAKE-A-SHAPE, DDIM]. Our results from WALA transfer to MAKE-A-SHAPE: As we move from a healthy to an unhealthy run, we observe that the baseline case shows a smooth rise in spectral entropy and a sudden jump in connectivity. Patching our $\mathbf{Y}$ keeps the spectral entropy at healthy levels and preserves connectivity. This behavior is consistent across diffusion seeds.

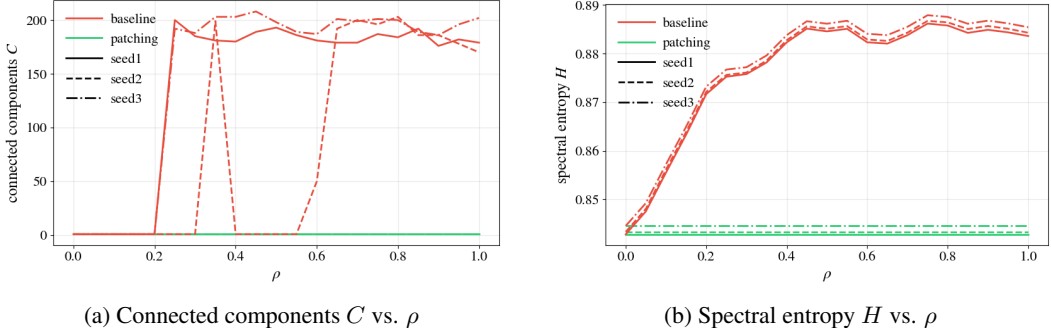

(a) Connected components $C$ vs. $\rho$    (b) Spectral entropy $H$ vs. $\rho$

Figure 10: [MAKE-A-SHAPE, DDPM]. Our results from [MAKE-A-SHAPE under DDIM sampling transfer to WALA under DDPM sampling.

## B.3 DATASET EVALUATION

This section provides (i) evidence that *meltdown* exists across a variety of shapes, i.e., across the GSO and SimJEB corpora, and diffusion transformers, i.e., WALA and MAKE-A-SHAPE. Furthermore it details our setup to evaluate our method `PowerRemap` on GSO (Downs et al., 2022) and SimJEB as well as the results of these evaluations.

**General.** We evaluate `PowerRemap` on the WALA and MAKE-A-SHAPE architectures, using DDIM sampling Song et al. (2021). For each object, we load the corresponding mesh as the ground-truth surface $\mathcal{S} \subset \mathbb{R}^3$, center it and scale it to the unit cube.

**Find meltdown.** We reuse the notation of §2. Given a mesh $\mathcal{S}$ and generator $G$, we first determine a sparse point budget by searching the smallest $N$ over a grid for which a Poisson-disk sample $A = \{a_i\}_{i=1}^N \subset \mathcal{S}$ yields a healthy output $C(G(A)) = 1$. We then define a surface-constrained *meltdown path* by jittering $A$ and projecting back to $\mathcal{S}$ to obtain $B = \{b_i\}_{i=1}^N$, and interpolate on-manifold

$$\mathcal{P}_\rho = \Pi_{\mathcal{S}}\big((1-\rho)A + \rho B\big), \quad \rho \in [0,1],$$

where $\Pi_{\mathcal{S}}$ is nearest-point projection. With a fixed random seed (reseeded before every inference), we sweep $\rho$ on a geometric grid to bracket a jump in connectivity, then refine by bisection to the smallest $\varepsilon$ such that $C(G(\mathcal{P}_\varepsilon)) \gg 1$.

---

**Algorithm 2** Adversarial meltdown search

---

**Require:** surface $\mathcal{S}$, generator $G$, component counter $C(\cdot)$, seed $s = 0$
1: normalize $\mathcal{S}$; find smallest $N$ s.t. $A \sim \text{Poisson}(\mathcal{S}, N)$ gives $C(G(A))=1$ (reseed $s$)
2: $B \leftarrow \Pi_{\mathcal{S}}(A + \xi)$                                                ▷ jitter & project
3: sweep $\rho$ on a geometric grid; find $\rho_{\text{lo}} < \rho_{\text{hi}}$ with $C(\rho_{\text{lo}})=1$, $C(\rho_{\text{hi}})>1$
4: $\varepsilon \leftarrow \text{bisection}(\rho_{\text{lo}}, \rho_{\text{hi}})$ with reseeding to $s$
5: **return** $(N, \varepsilon, C_0 = C(G(A)), C_\varepsilon = C(G(\mathcal{P}_\varepsilon)))$

---

**Evaluate `PowerRemap`.** The task of reconstructing a global surface from a sparse point cloud has only two possible outcomes: success or failure. Thus, we assess the effectiveness `PowerRemap` on GSO by counting the number of times it succeeded in reducing $C_\varepsilon$ to 1, i.e., turning a speckle into a shape. Hence, we treat each shape as a Bernoulli trial under our adverserial search (Algorithm 2). Each trail has an outcome $p \in \{0, 1\}$, where $p = 1$ iff the reconstruction meets the criterion $C_\varepsilon = 1$; otherwise $p = 0$. We first identify baseline failures as those with $C_{0,\text{baseline}} = 1$ and $C_{\varepsilon,\text{baseline}} > 1$. We then apply `PowerRemap` only to these failures and count a remedy when $C_{\varepsilon,\text{PowerRemap}} = 1$.

### B.3.1 GSO

This section provides a quantitative evaluation of the *meltdown* phenomenon on the GSO dataset (Downs et al., 2022) and assesses the effectiveness of `PowerRemap` as a mitigation strategy. SimJEB is a curated benchmark of 381 3D jet-engine bracket CAD models that was *not* included in the training data of either WALA or MAKE-A-SHAPE. All results in this section are obtained by applying the protocol described in Appendix B.3 to the SimJEB dataset.

**Evaluate `PowerRemap` for global $\gamma = 100$.** For WALA, we found meltdown in 926/1,030 (89.9%) shapes. Our method `PowerRemap` remedies failure in 910/926 (98.3%) for $\gamma = 100$. Table 3 depicts the performance of our method across all shape categories. For MAKE-A-SHAPE, we found meltdown in 910/1,030 (88.9%) shapes. For a *global* $\gamma = 100$, our method `PowerRemap` remedies failure in 92/910 (10.1%) cases ( cf. Table 4).

**Evaluate `PowerRemap` for adpative-$\gamma$ strategy.** We further investigate whether *per-instance tuning* of $\gamma$ can recover meltdown cases that are not resolved by a single global setting. Specifically, we consider 130 GSO shapes for which meltdown persists under the global choice $\gamma = 100$ (cf.

Table 4) and re-evaluate `PowerRemap` on these shapes over the grid

$$\gamma \in \{1.05, 1.1, 1.15, 1.2, 1.25, 1.3, 1.35, 1.4, 1.5, 2\}.$$

For 110 out of 130 shapes (84.6%), we find at least one value of $\gamma$ in this range that successfully remedies meltdown. Among the rescued cases, the median effective value of $\gamma$ is 1.10 with a standard deviation of 0.13.

| Category | #Shapes | Meltdown | Meltdown (%) | PR Success | PR@Melt |
|---|---|---|---|---|---|
| **All** | 1030 | **926** | **89.9** | 910 | **98.3** |
| Shoe | 254 | 247 | 97.2 | 246 | 99.6 |
| Consumer Goods | 248 | 242 | 97.6 | 240 | 99.2 |
| Unknown | 216 | 191 | 88.4 | 183 | 95.8 |
| Toys | 147 | 89 | 60.5 | 85 | 95.5 |
| Bottles and Cans and Cups | 53 | 53 | 100.0 | 53 | 100.0 |
| Bag | 28 | 26 | 92.9 | 26 | 100.0 |
| Media Cases | 21 | 21 | 100.0 | 21 | 100.0 |
| Action Figures | 17 | 16 | 94.1 | 15 | 93.8 |
| Board Games | 17 | 16 | 94.1 | 16 | 100.0 |
| Legos | 10 | 6 | 60.0 | 6 | 100.0 |
| Headphones | 4 | 4 | 100.0 | 4 | 100.0 |
| Keyboard | 4 | 4 | 100.0 | 4 | 100.0 |
| Mouse | 4 | 4 | 100.0 | 4 | 100.0 |
| Stuffed Toys | 3 | 3 | 100.0 | 3 | 100.0 |
| Hat | 2 | 2 | 100.0 | 2 | 100.0 |
| Camera | 1 | 1 | 100.0 | 1 | 100.0 |
| Car Seat | 1 | 1 | 100.0 | 1 | 100.0 |

Table 3: [WALA (Sanghi et al., 2024)] Evaluation of `PowerRemap` on GSO for $\gamma = 100$. *Meltdown* counts shapes where the baseline generator yields a single component on the unperturbed input but multiple components after a small, on-surface perturbation ($C_0{=}1$, $C_\varepsilon{>}1$). *Meltdown (%)* is Meltdown divided by #Shapes. *PR Success* counts—among meltdown shapes only—cases where `PowerRemap` restores a single component ($C_{\varepsilon,\texttt{PowerRemap}} = 1$). *PR@Melt* is the success rate on meltdown shapes (PR Success / Meltdown, shown in %).

| Category | #Shapes | Meltdown | Meltdown (%) | PR Success | PR@Melt (%) |
|---|---|---|---|---|---|
| Global | 1030 | 910 | 88.3 | 92 | 10.1 |
| Consumer Goods | 248 | 235 | 94.8 | 26 | 11.1 |
| Shoe | 254 | 234 | 92.1 | 17 | 7.3 |
| **Unknown** | 216 | 198 | 91.7 | 24 | 12.1 |
| Toys | 147 | 83 | 56.5 | 9 | 10.8 |
| Bottles and Cans and Cups | 53 | 50 | 94.3 | 1 | 2.0 |
| **Bag** | 28 | 28 | 100.0 | 8 | 28.6 |
| **Media Cases** | 21 | 21 | 100.0 | 3 | 14.3 |
| Board Games | 17 | 17 | 100.0 | 0 | 0.0 |
| Action Figures | 17 | 15 | 88.2 | 1 | 6.7 |
| Legos | 10 | 6 | 60.0 | 1 | 16.7 |
| Mouse | 4 | 4 | 100.0 | 1 | 25.0 |
| Headphones | 4 | 3 | 75.0 | 0 | 0.0 |
| Stuffed Toys | 3 | 3 | 100.0 | 1 | 33.3 |
| Keyboard | 4 | 2 | 50.0 | 0 | 0.0 |
| Hat | 2 | 2 | 100.0 | 0 | 0.0 |
| Camera | 1 | 1 | 100.0 | 0 | 0.0 |
| Car Seat | 1 | 1 | 100.0 | 0 | 0.0 |
| Macro avg | – | – | 88.4 | – | 9.9 |

Table 4: [MAKE-A-SHAPE (Hui et al., 2024)] Evaluation of `PowerRemap` on GSO for $\gamma = 100$. *Meltdown* counts shapes where the baseline generator yields a single component on the unperturbed input but multiple components after a small, on-surface perturbation ($C_0{=}1$, $C_\varepsilon{>}1$). *Meltdown (%)* is Meltdown divided by #Shapes. *PR Success* counts—among meltdown shapes only—cases where `PowerRemap` restores a single component ($C_{\varepsilon,\texttt{PowerRemap}} = 1$). *PR@Melt* is the success rate on meltdown shapes (PR Success / Meltdown, shown in %).

Table 5: Category-wise evaluation of `PowerRemap` on the 130-shape MAS subset (Top-3 Categories). Our method stabilizes failures in 84.6% of cases.

| Category | Shapes | Meltdown occurs [%] | `PowerRemap` rescues [%] |
|---|---|---|---|
| Consumer goods | 60 | 100.0 | 90.0 |
| Bottles, cans & cups | 23 | 100.0 | 95.7 |
| Unknown | 18 | 100.0 | 83.3 |
| Other | 29 | 100.0 | 65.5 |
| **Total** | **130** | **100.0** | **84.6** |

### B.4   SIMJEB

This section provides a quantitative evaluation of the *meltdown* phenomenon on the SimJEB dataset (Whalen et al., 2021) and assesses the effectiveness of `PowerRemap` as a mitigation strategy. Sim-JEB is a curated benchmark of 381 3D jet-engine bracket CAD models that was *not* included in the training data of either WALA or MAKE-A-SHAPE. All results in this section are obtained by applying the protocol described in Appendix B.3 to the SimJEB dataset.

**Meltdown**   We detect meltdown in 352 out of 381 shapes (92.4%) generated by WALA and in 363 out of 381 shapes (95.3%) generated by MAKE-A-SHAPE. The distribution of meltdown across Sim-JEB shape categories is reported in Table 6 for WALA and in Table 7 for MAKE-A-SHAPE. These results indicate that meltdown is pervasive across categories and affects both generative pipelines at similar rates.

Table 6: Category-wise evaluation of meltdown on SimJEB (WALA).

| Category | Shapes | Meltdowns | Meltdown [%] | Avg. areal density |
|---|---|---|---|---|
| Arch | 37 | 33 | 89.2 | 1.225e-02 |
| Beam | 46 | 46 | 100.0 | 1.065e-02 |
| Block | 99 | 87 | 87.9 | 8.413e-03 |
| Butterfly | 43 | 40 | 93.0 | 1.282e-02 |
| Flat | 147 | 137 | 93.2 | 9.785e-03 |
| Other | 9 | 9 | 100.0 | 1.400e-02 |
| **Total** | **381** | **352** | **92.4** | 1.024e-02 |

Table 7: Category-wise evaluation of meltdown on SimJEB (MAKE-A-SHAPE).

| Category | Shapes | Meltdowns | Meltdown [%] | Avg. areal density |
|---|---|---|---|---|
| Arch | 37 | 36 | 97.3 | 1.920e-02 |
| Beam | 46 | 46 | 100.0 | 1.856e-02 |
| Block | 99 | 87 | 87.9 | 1.371e-02 |
| Butterfly | 43 | 42 | 97.7 | 1.491e-02 |
| Flat | 147 | 143 | 97.3 | 1.370e-02 |
| Other | 9 | 9 | 100.0 | 8.706e-03 |
| **Total** | **381** | **363** | **95.3** | 1.488e-02 |

**Evaluate `PowerRemap`**   We evaluate `PowerRemap` on the meltdown cases identified. For WALA, we apply `PowerRemap` with a fixed hyperparameter $\gamma = 100$ to each meltdown instance and measure the fraction of cases in which the failure is resolved. Under this setting, `PowerRemap` remedies 97.7% of meltdown cases; the category-wise breakdown is reported in Table 8. For MAKE-A-SHAPE, we consider a category-representative subset of 30 SimJEB shapes and evaluate `PowerRemap` over the hyperparameter grid $\gamma \in \{1.05, 1.1, 1.15, 1.2, 1.25, 1.3, 1.35, 1.4, 1.5, 2\}$. Across this range, `PowerRemap` successfully remedies meltdown in 83.3% of the evaluated cases with a median effective $\gamma$ of 1.05 (standard deviation 0.063).

Table 8: Category-wise evaluation of `PowerRemap` on SimJEB (WALA).

| Category | Shapes | Meltdowns | PowerRemap rescues | PowerRemap rescues [%] |
|---|---|---|---|---|
| Arch | 37 | 3 | 33 | 100.0 |
| Beam | 46 | 46 2 | 46 | 100.0 |
| Block | 99 | 87 3 | 85 | 97.7 |
| Butterfly | 43 | 40 | 38 | 95.0 |
| Flat | 147 | 137 | 134 | 97.8 |
| Other | 9 | 9 | 8 | 88.9 |
| **Total** | 381 | 352 | 344 | **97.7** |

Table 9: Category-wise evaluation of `PowerRemap` on SimJEB (MAKE-A-SHAPE).

| Category | Shapes | Meltdowns | PowerRemap rescues | PowerRemap rescues [%] |
|---|---|---|---|---|
| Arch | 2 | 2 | 1 | 50.0 |
| Beam | 4 | 4 | 3 | 75.0 |
| Block | 9 | 9 | 9 | 100.0 |
| Butterfly | 3 | 3 | 2 | 66.7 |
| Flat | 11 | 11 | 10 | 90.9 |
| Other | 1 | 1 | 0 | 0.0 |
| **Total** | 30 | 30 | 25 | **83.3** |

## C PowerRemap

*Proof of Proposition 1.* Let $\sigma_i \geq 0$ be the singular values of $\mathbf{Y}$ and set $z_i := \sigma_i^2$ (unnormalized directional energies). The baseline normalized spectrum is $p_i := z_i / \sum_j z_j$, with entropy $H(\mathbf{Y}) = -\sum_i p_i \log p_i$.

By definition, $\sigma_i' = \sigma_{\max}(\sigma_i/\sigma_{\max})^\gamma$, hence $(\sigma_i')^2 = \sigma_{\max}^{2-2\gamma}\sigma_i^{2\gamma} = \kappa z_i^\gamma$ with a common $\kappa > 0$ that cancels upon normalization. Therefore the post-intervention normalized spectrum is

$$p_i^{(\gamma)} = \frac{z_i^\gamma}{\sum_j z_j^\gamma}.$$

We note that indices with $z_i = 0$ keep $p_i^{(\gamma)} = 0$ for all $\gamma > 0$ and can be excluded without loss. Write $s_i := \log z_i$ (for the retained indices). Then

$$p_i^{(\gamma)} = \frac{e^{\gamma s_i}}{\sum_j e^{\gamma s_j}} \qquad \text{and} \qquad \phi(\gamma) := \log \sum_j e^{\gamma s_j}.$$

Thus $\log p_i^{(\gamma)} = \gamma s_i - \phi(\gamma)$ and the spectral entropy after the intervention is

$$H(\gamma) := -\sum_i p_i^{(\gamma)} \log p_i^{(\gamma)} = \phi(\gamma) - \gamma \sum_i p_i^{(\gamma)} s_i. \tag{10}$$

Using the standard identity for the softmax measure,

$$\phi'(\gamma) = \frac{\sum_i s_i e^{\gamma s_i}}{\sum_j e^{\gamma s_j}} = \sum_i p_i^{(\gamma)} s_i =: \mathbb{E}_{p^{(\gamma)}}[s],$$

equation 10 simplifies to

$$H(\gamma) = \phi(\gamma) - \gamma\,\phi'(\gamma).$$

Differentiating once gives

$$H'(\gamma) = \phi'(\gamma) - \big(\phi'(\gamma) + \gamma\,\phi''(\gamma)\big) = -\gamma\,\phi''(\gamma).$$

It remains to identify $\phi''(\gamma)$. A direct computation shows

$$\frac{d}{d\gamma} p_i^{(\gamma)} = p_i^{(\gamma)}\big(s_i - \mathbb{E}_{p^{(\gamma)}}[s]\big), \quad \text{hence} \quad \phi''(\gamma) = \frac{d}{d\gamma}\mathbb{E}_{p^{(\gamma)}}[s] = \sum_i s_i \frac{d}{d\gamma} p_i^{(\gamma)} = \mathrm{Var}_{p^{(\gamma)}}(s) \geq 0.$$

Therefore

$$H'(\gamma) = -\gamma\,\mathrm{Var}_{p^{(\gamma)}}(s) \leq 0,$$

with equality iff all $\sigma_i > 0$ are equal.

We conclude that $H(\gamma)$ is nonincreasing in $\gamma$; in particular, for any $\gamma > 1$,

$$H\big(\texttt{PowerRemap}(\mathbf{Y})\big) = H(\gamma) \leq H(1) = H(\mathbf{Y}).$$

$\square$

# D  USE OF LARGE LANGUAGE MODELS (LLMS)

To a limited extent, we used LLMs to **aid or polish writing**, i.e., in some instances, we used LLMs to reformulate sentences.

# E DENSITY

We quantify the sparsity of an input point cloud of size $N$ by its *areal density*

$$\eta = \frac{N}{A_\mathcal{S}}, \tag{11}$$

where $A_\mathcal{S}$ denotes the surface area of the underlying surface $\mathcal{S}$. Higher values of $\eta$ correspond to denser samplings of $\mathcal{S}$.

In Figure 11, we examine how the prevalence of meltdown depends on $\eta$ for SimJEB shape 492. For each target areal density, we run Algorithm 2 (parameterized by $\eta$) for 10 independent trials and record how often a meltdown configuration is identified. We observe that meltdown is particularly frequent in the low-$\eta$ regime.

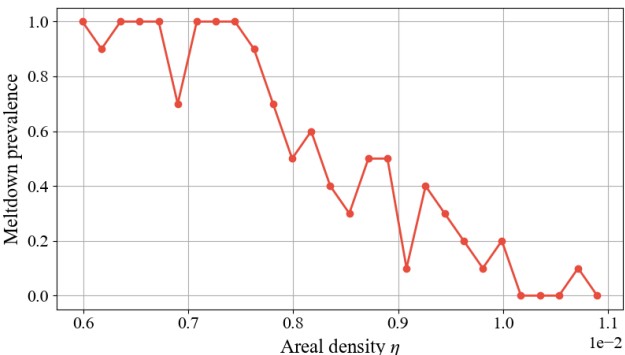

Figure 11: Incidence of meltdown on SimJEB shape 492 as a function of areal density $\eta$. Meltdown events are especially common for low areal densities, underscoring the difficulty of robust surface reconstruction from sparse point clouds.

# F   ACTIVATION PATCHING

The decision to restrict the search-space for activation-patching in Section 3.2 to cross-attention activations is motivated by prior work (Surkov et al., 2025; Tang et al., 2022; Tinaz et al., 2025).   As depicted in Figure 12 extending the search for WALA in [1] to self-attention $\mathring{\mathbf{Y}}$, residual $\overset{\times}{\mathbf{R}}$, and MLP activations $\bar{\mathbf{Y}}$ reveals that the early cross-attention block identified in Figure 2 is the source of meltdown and other components merely carry the signal downstream.  Additionally, the authors found in their experiments cross-attention to be the only interpretable component (cf. Figure 14 and Figure 15b).

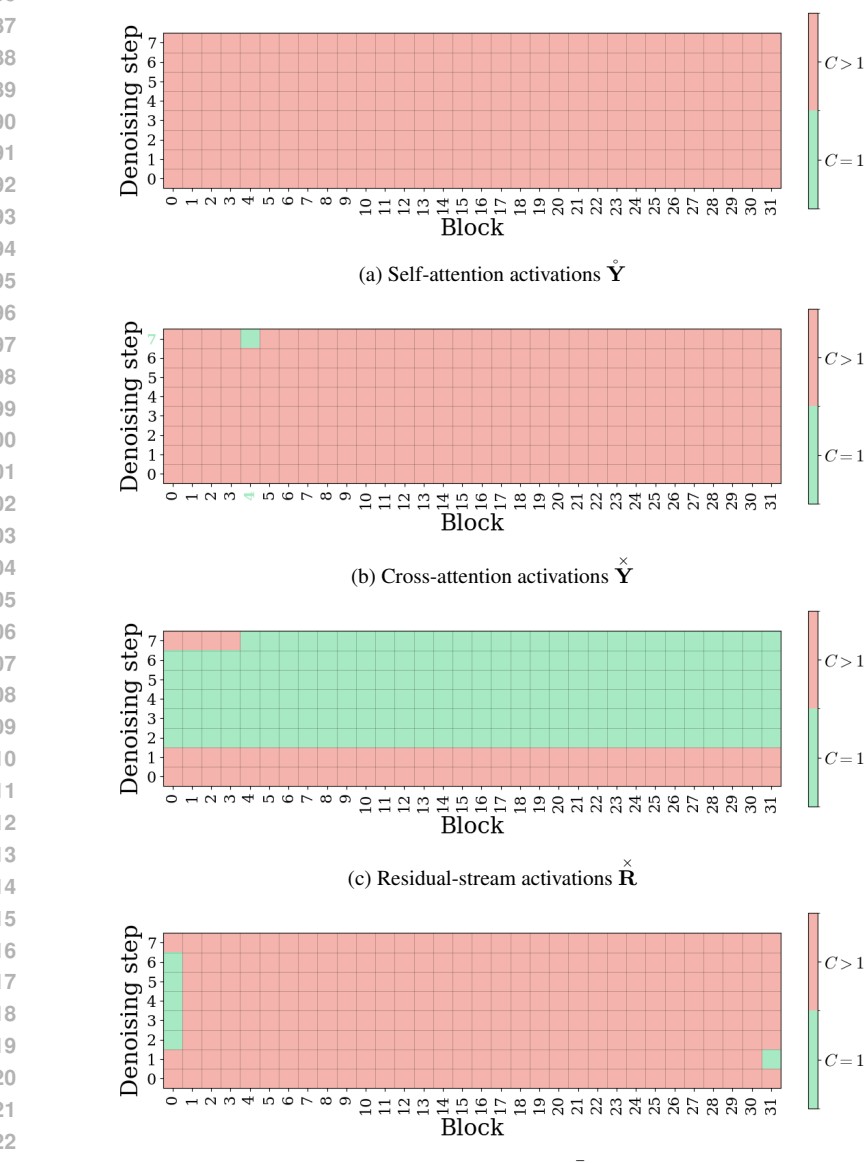

(a) Self-attention activations $\mathring{\mathbf{Y}}$

(b) Cross-attention activations $\overset{\times}{\mathbf{Y}}$

(c) Residual-stream activations $\overset{\times}{\mathbf{R}}$

(d) MLP activations $\bar{\mathbf{Y}}$

Figure 12: Results for depth-time grid search for self-attention, cross-attention, residual-stream and MLP activations.

---

[1]Results are averaged over three trials using the setup described in Section 2.

## G MORE DATAPOINTS

In this section, we provide additional evidence that the patterns observed in Section 3.2-3.3 generalize when evaluated on more data points and random seeds . Figure 13 and Figure 14 show that the behavior transfers to diverse shapes from the GSO (Downs et al., 2022) and SimJEB (Whalen et al., 2021) corpora as well as diffusion seeds for the WALA model. Figure 15 shows that the average behavior over a population of 150 diffusion seeds for SimJEB shape 492 is consistent with the observations reported in Section 3.3 for the WALA model.

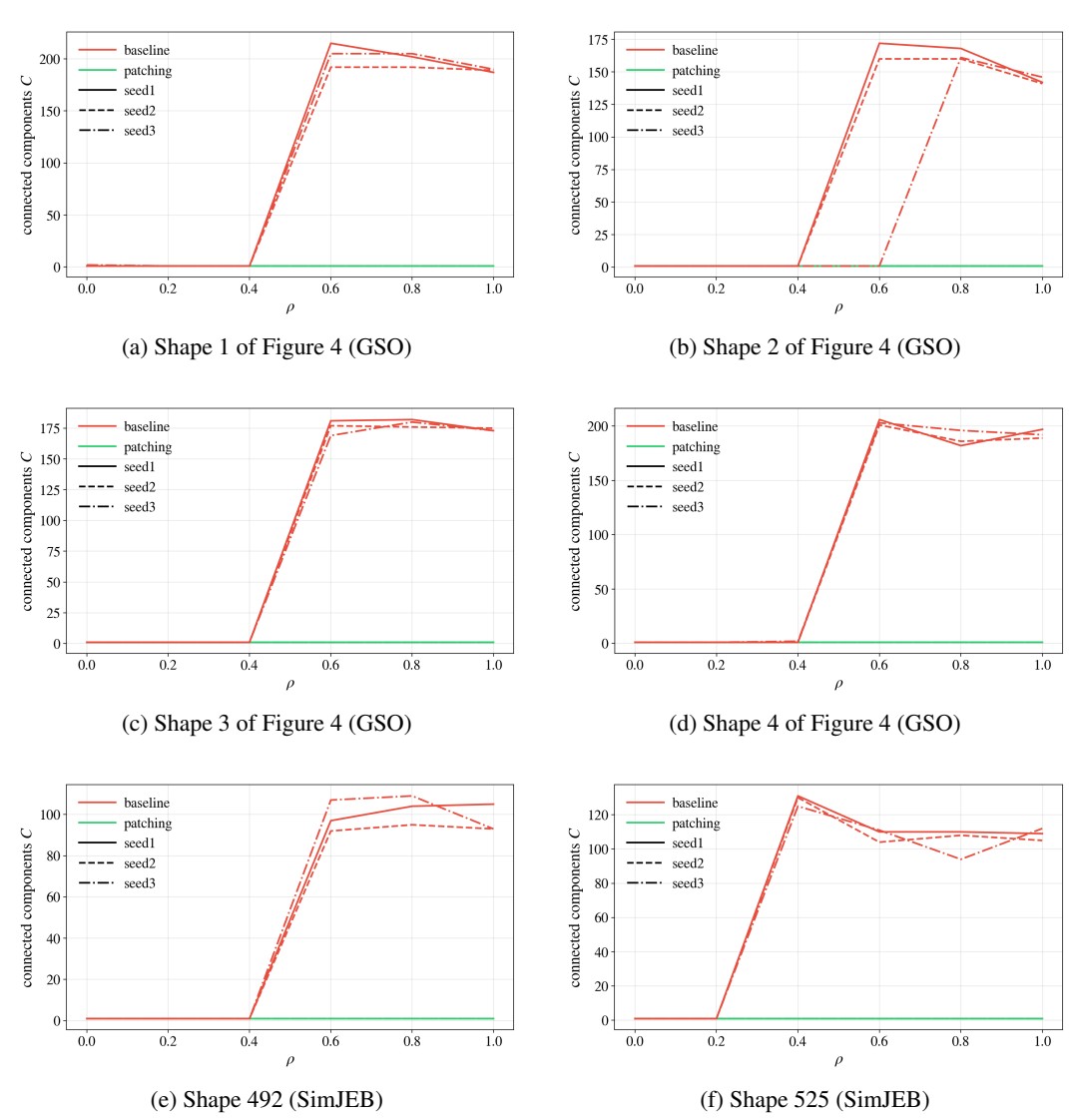

(a) Shape 1 of Figure 4 (GSO)

(b) Shape 2 of Figure 4 (GSO)

(c) Shape 3 of Figure 4 (GSO)

(d) Shape 4 of Figure 4 (GSO)

(e) Shape 492 (SimJEB)

(f) Shape 525 (SimJEB)

Figure 13: Connected components $C$ vs. $\rho$.

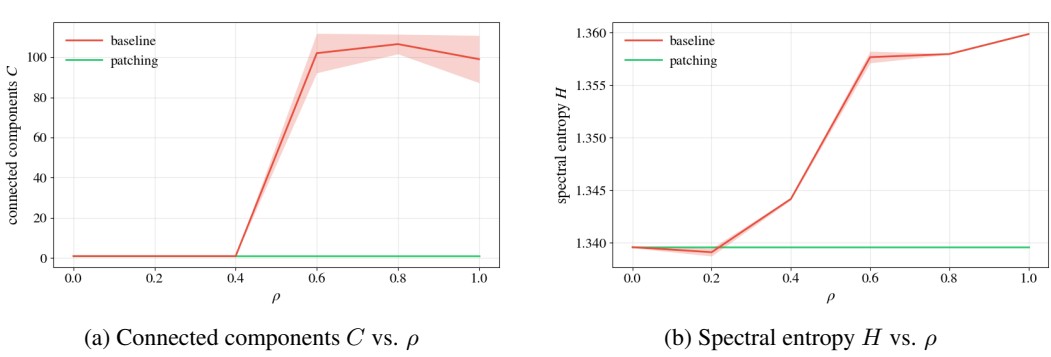

(a) Shape 1 of Figure 4 (GSO)

(b) Shape 2 of Figure 4 (GSO)

(c) Shape 3 of Figure 4 (GSO)

(d) Shape 4 of Figure 4 (GSO)

(e) Shape 492 (SimJEB)

(f) Shape 525 (SimJEB)

Figure 14: Spectral entropy $H$ vs. $\rho$.

(a) Connected components $C$ vs. $\rho$

(b) Spectral entropy $H$ vs. $\rho$

Figure 15: Patterns at population level for SimJEB shape 492, using 150 diffusion seeds.

# H  ADDITIONAL SPECTRAL METRICS

In this section, we analyze additional spectral metrics to assess their suitability as indicators of meltdown for the WALA model. In particular, Figure 13 reports the effective rank $r_{\text{eff}} = \exp(H)$ (cf. Definition 1) and Figure 17 the condition number $\kappa = \sigma_{\max}/\sigma_{\min}$ (cf. Definition 1) as alternatives to spectral entropy for a diverse set of shapes. We observe that the effective rank—which is a monotonic transformation of spectral entropy—provides an equally informative indicator of meltdown. By contrast, the condition number exhibits no apparent correlation with the failure phenomenon, suggesting that it is not a suitable diagnostic metric in this setting.

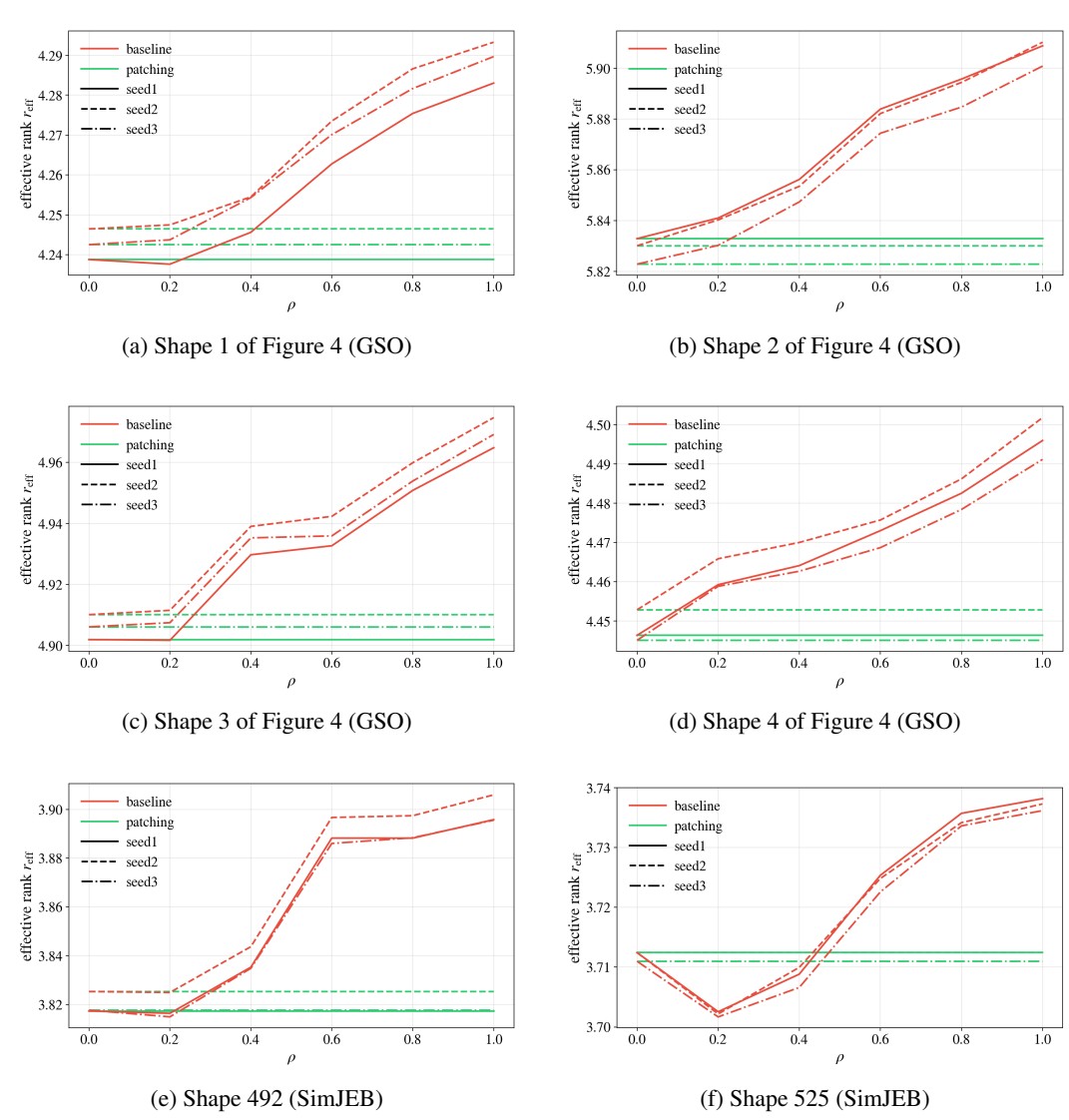

(a) Shape 1 of Figure 4 (GSO)

(b) Shape 2 of Figure 4 (GSO)

(c) Shape 3 of Figure 4 (GSO)

(d) Shape 4 of Figure 4 (GSO)

(e) Shape 492 (SimJEB)

(f) Shape 525 (SimJEB)

Figure 16: Effective rank $r_{\text{eff}}$ vs. $\rho$.

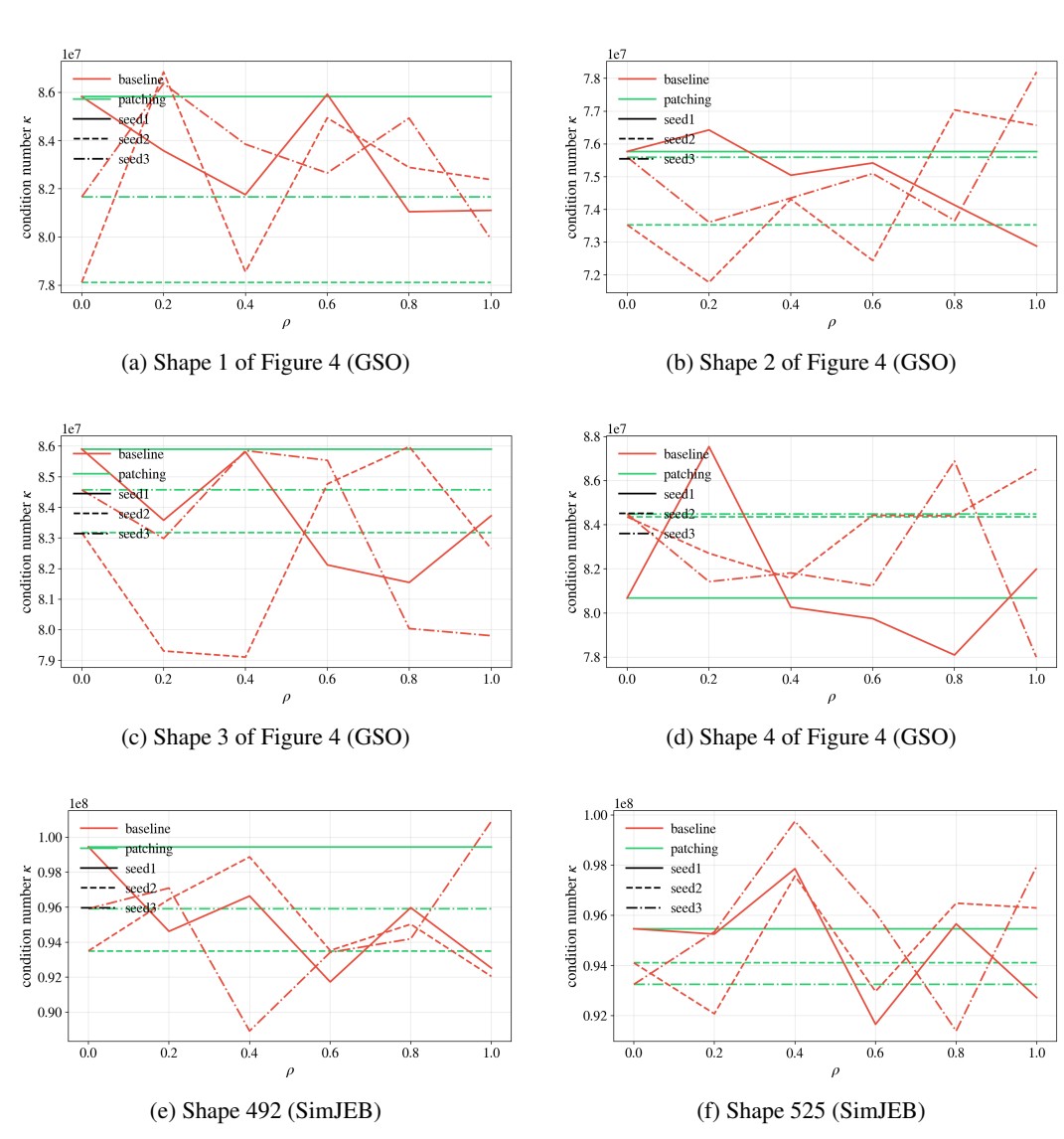

(a) Shape 1 of Figure 4 (GSO)

(b) Shape 2 of Figure 4 (GSO)

(c) Shape 3 of Figure 4 (GSO)

(d) Shape 4 of Figure 4 (GSO)

(e) Shape 492 (SimJEB)

(f) Shape 525 (SimJEB)

Figure 17: Condition number $\kappa$ vs. $\rho$.

# I  POWERREMAP EVALUATION

## I.1  MULTIPLE OBJECTS

We further assess whether the meltdown phenomenon and the effectiveness of `PowerRemap` extend beyond single-object inputs. Figure 18 provides a qualitative evaluation on a scene containing multiple objects for the WALA model. We observe that meltdown still occurs in this multi-object setting, while using `PowerRemap` reliably suppresses the failure and preserves a plausible reconstruction of all objects in the scene.

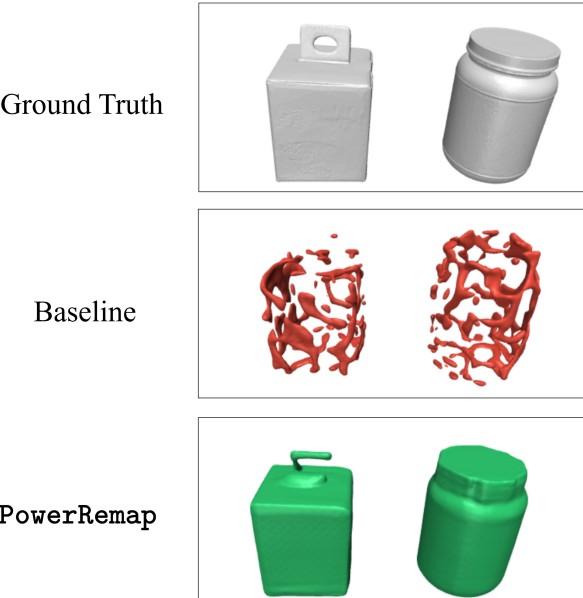

Figure 18: Qualitative evaluation of multi-object inputs. Meltdown persists in scenes with multiple objects, leading to severe degradation of the reconstruction, whereas `PowerRemap` effectively prevents this failure mode and yields a stable reconstruction of all objects.

## I.2 Non-meltdown cases

We empirically verify that `PowerRemap` does not interfere with non-meltdown runs . Specifically, we run Algorithm 2 on SimJEB (WALA) shape 492 for 10 independent random seeds and select a configuration with $C_0 = 1$ (a single connected component). We then apply `PowerRemap` to this configuration over the hyperparameter grid $\gamma \in \{2, 5, 10, 100\}$. As can be seen in Table 10, the reconstructed surface retains $C_0 = 1$ in all cases and we do not observe any change in the final topology. The results in Table 10 indicate that `PowerRemap` is effectively topologically neutral on this non-meltdown instance.

Table 10: Evaluation of `PowerRemap` on a non-meltdown SimJEB shape (shape 492). Across 10 random seeds and $\gamma \in \{2, 5, 10, 100\}$, `PowerRemap` preserves the original topology ($C_0 = 1$) in all cases.

| Shapes | Seeds | Meltdown occurs [%] | Topology preserved [%] |
|---|---|---|---|
| SimJEB 492 | 10 | 0.0 | 100.0 |
| **Total** | **10** | **0.0** | **100.0** |

## I.3 CHOICE OF $\gamma$ TO REMEDY MELTDOWN

We empirically investigate the influence of the `PowerRemap` strength $\gamma$ on reconstruction connectivity.

**WALA**  We run Algorithm 2 on SimJEB (WALA) shape 492 for 10 independent random seeds and select the meltdown configuration with $C_{\varepsilon} > 1$. We then apply `PowerRemap` to this configuration over the hyperparameter grid $\gamma \in \{1, 2, 5, 10, 100\}$, where $\gamma = 1$ denotes the identity mapping. As can be seen in 19, our `PowerRemap` method achieves a high success rate for $\gamma > 2$.

**MAKE-A-SHAPE**  For MAKE-A-SHAPE, we investigate the distribution of `PowerRemap` strenghts $\gamma$ for the 130-shape subset of GSO as discussed in Table 5. We find that $\gamma$ values around 1.05 are effective to remedy meltdown.

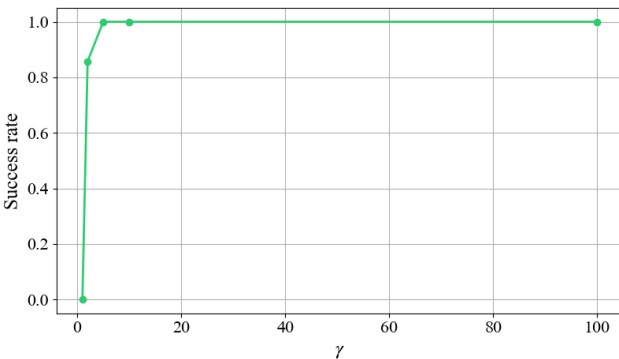

Figure 19: For the `WaLa` model, we find that a `PowerRemap` strength of $\gamma > 2$ remedies meltdown.

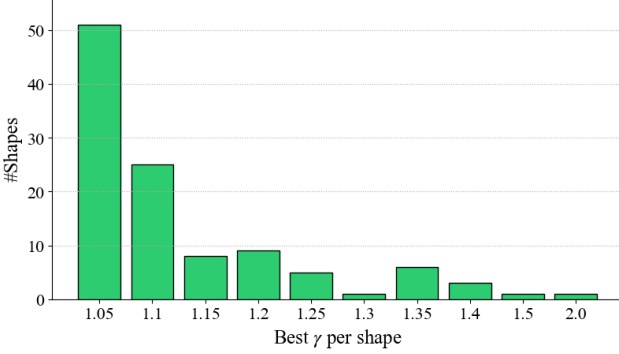

Figure 20: For the 130-shape subset in Table 5 (MAKE-A-SHAPE), we find that $\gamma$ values around 1.05 are effective to remedy meltdown.