# OpenReview forum: "From Circuits to Dynamics: Understanding and Stabilizing Failure in 3D Diffusion Transformers"
_ICLR.cc/2026/Conference — Submitted to ICLR 2026_

### Official Review · Reviewer_5ZNt · 2025-10-18

**Soundness:** 3
**Presentation:** 3
**Contribution:** 3
**Rating:** 6
**Confidence:** 3

**Summary:**

This paper reveals and deeply analyzes a catastrophic failure phenomenon that exists in the current most advanced 3D diffusion Transformers when performing surface reconstruction from sparse point clouds, which the authors call meltdown. Specifically, a tiny perturbation located on the surface of the input point cloud may cause the model output to melt from a complete single shape into a large number of unconnected fragments.

The author first demonstrated the existence of this phenomenon and found that the spectral entropy of this key activation could serve as a measurable proxy indicator for the occurrence of the collapse. Based on this insight, the author proposed a simple yet effective test-time intervention method called PowerRemap. The author provides an explanation for the meltdown phenomenon from the perspective of diffusion dynamics, linking it to the symmetry-breaking bifurcations of the potential energy landscape during the reverse diffusion process, thereby associating the discoveries at the circuit level with the deeper theoretical framework of generative models.

**Strengths:**

This is a high-quality and fascinating study. It combines solid experiments, ingenious interpretability analysis, effective solutions and profound theoretical connections, which is of great significance for improving the robustness and interpretability of diffusion models, especially 3D generative models.

**Weaknesses:**

The paper put forward the "consensus" hypothesis, but this has not yet been strictly verified.

Currently, the intensity parameter γ is a manually set global value (γ=100), which is not optimal and also limits the robustness of the method.

**Questions:**

1. Has the author ever attempted an adaptive γ selection strategy?

2. Please discuss in more detail the reasons for the poor performance on Make-A-Shape.

3. In the main text or appendix, the measurement criteria for point cloud sparsity should be more clearly stated. Can the average density of the point cloud relative to the surface area of the object or the sampling distance of the farthest point be provided? This helps readers better understand the challenge of the problem setting.

4. There is still some speculation about the mechanism explanation for "why does a decreased spectral entropy reduce invalid outputs?". It is suggested that the author deepen this explanation. For instance, can we analyze whether there are any changes in the alignment or consistency of the outputs from different attention heads before and after the PowerRemap intervention? Or, can the assumption that "the first singular vector represents the consensus feature of multiple heads" be verified through the analysis of singular vectors? Even a preliminary analysis can significantly enhance the persuasiveness of the argument.

---

> ### Author Response · Authors · 2025-11-22
>
> We thank the reviewer for their careful review and for the  valuable suggestions to improve our work. We appreciate that the reviewer finds our study high-quality and fascinating, and we are happy to provide detailed answers to the comments below.
>
> ## Consensus hypothesis (W1)
>
> We thank the reviewer for this thoughtful comment and for pushing us to be clearer. Our statement that “the first singular vector represents the consensus feature of multiple heads”: when we stack all head outputs into matrix $\boldsymbol{Y}$, the leading singular vector is by construction the direction that maximizes the summed squared projection of all heads, i.e., the dominant shared feature direction. PowerRemap then reduces spectral entropy by reweighting the singular values so that this dominant shared direction becomes relatively stronger, which we interpret as enforcing a stronger “consensus” across heads. We have adapted Section 6 in the manuscript accordingly.
>
>
> ## Adaptive $\gamma$-strategy (W2)
>
> Thank you for raising this valuable question.
> Indeed we observed in our newly added experiments that the poor performance of MAS on GSO was due to a suboptimal selection of $\gamma=100$. We now performed a grid-search over different values for $\gamma$ for MAS for a subset of objects of the GSO and SimJEB datasets and observed an optimal value of $\gamma=1.05 \pm 0.13$ and $\gamma=1.05 \pm 0.06$, respectively. This finding indicates that the choice of the optimal $\gamma$ is mostly dependent on the model as the variance in $\gamma$ is rather small for different shapes. In the future we envision to find a more sophisticated manner to infer the optimal value of $\gamma$ for different models.
>
> ## Questions
>
> 1. See W2
> 2. See W2
>
> 3. __Point cloud sparsity__
>
> In line with the recommendations from the reviwer (as well as reviewer 9Hbz), we have augmented our results to include measurements of areal density. Accordingly, we have added areal-density summary statistics to Tables 1 and 2. Moreover, Appendix E now provides an analysis of meltdown prevalence as a function of areal density. Figure 11 shows that meltdown is particularly frequent in regimes of low areal density.
>
> 4. See W1
>
> ## References
>
> [1] Laura Downs, Anthony Francis, Nate Koenig, Brandon Kinman, Ryan Hickman, Krista Reymann,Thomas B. McHugh, and Vincent Vanhoucke. Google scanned objects: A high-quality dataset of 3D scanned household items, 2022. URL: https://arxiv.org/abs/2204.11918.
>
> [2] E. Whalen, A. Beyene, and C. Mueller. SimJEB: Simulated Jet Engine Bracket Dataset. Computer Graphics Forum, 40(5): 9–17, August 2021. ISSN 1467-8659. doi:10.1111/cgf.14353.

---

### Official Review · Reviewer_9Hbz · 2025-10-28

**Soundness:** 3
**Presentation:** 3
**Contribution:** 3
**Rating:** 6
**Confidence:** 3

**Summary:**

The paper studies a common catastrophic mode of faillure for 3D diffusion model in the task of surfuace completion from sparse point clouds . They name it as melt-down and performs activation-patching to localize the failure position. They leverages the singular-value spectrum of the located activation module to serve as a proxy for the failure mode. To tackle this issue, they proposes PowerRemap module to adjust the singular value as a test-time module. Experiments are done on GSO to illustrate the idea.

**Strengths:**

1. The meltdown phenomenon is a common issue in 3D diffusion models for shape completion and worth investigating.

2. The finding that a single cross-attention module is primarily responsible for the observed failure is particularly interesting and provides useful insight into the model’s internal behavior.

3. The discussion on diffusion dynamics is interesting and contributes to a better conceptual understanding of diffusion behavior.

**Weaknesses:**

1. The experiments are insufficient. The observed meltdown failure is likely to depend strongly on the density of the input point cloud, yet this factor is neither analyzed nor explicitly specified in the experiments. In addition, all experiments are conducted solely on the GSO dataset, which limits the generality of the conclusions. Including results on at least one additional dataset would significantly strengthen the empirical support for the proposed theory.

2. In Fig. 3, the trend of connectivity C does not fully align with that of H. Specifically, C rises sharply and reaches its maximum around ρ=0.4, then slightly decreases, whereas H continues to increase. This raises questions about the claimed relationship between H and C— why would a decreasing H correspond to improved connectivity? As presented, the experiment is not sufficiently convincing and requires clearer explanation or additional analysis.

3. The method appears to assume that the input point cloud is clean and contains only a single object. The definition of “healthy” results seems to rely on this restricted input condition. It is unclear how the approach would perform when the input represents a complete scene or includes significant sensor noise. Moreover, the theoretical formulation seems to inherently produce a single mesh, regardless of the input’s complexity or content.

**Questions:**

Please refer to weaknesses. Although the experimental evaluation is somewhat limited and certain analyses are not entirely convincing, the findings are novel and valuable. I would be glad to see this work published if the authors can provide additional experiments to strengthen the empirical validation.

---

> ### Author Response · Authors · 2025-11-22
>
> We thank the reviewer for their careful review and the many valuable suggestions for improving our work. We appreciate the assesment that our findings are novel and valuable, and are happy to provide additional experiments to strengthen the experimental validation. We hope the additional results address the concerns.
>
> ## Insufficient experiments (W1)
>
> We agree that our experimental results in its current form are limited. To provide evidence that the meltdown phenomenon generalizes beyond Google Scanned Objects (GSO) [1], we repeat our experiments on another dataset, namely SimJEB [2]. SimJEB is a curated benchmark consisting of 381 3D jet-engine bracket CAD models, and it was not included in the training data for either WALA or MAKE-A-SHAPE (MAS). We observe meltdown on SimJEB in 352 out of 381 shapes (92.4%) for WALA (see Table 1) and in 363 out of 381 shapes (95.3%) for MAS (see Table 7). We therefore conclude that the identified meltdown phenomenon exists across diverse shapes beyond GSO.
>
> We report results for WALA on SimJEB in the table below and also added it to Table 1 in our revised version. In addition to success rates we include measurements of areal density for each object category.
>
> Category | Shapes | Meltdowns | PR Rescues [%] | Density
> --- | --- | --- | --- | --- |
> Arch | 37 | 33 | 100.0 | 1.225e-02
> Beam | 46 | 46 | 100.0 | 1.065e-02
> Block | 99 | 87 | 97.7 | 8.413e-03
> Butterfly | 43 | 40 | 95.0 | 1.282e-02
> Flat | 147 | 137 | 97.8 | 9.785e-03
> Other | 9 | 9 | 88.9 | 1.400e-02
> Total | 381 | 352 | 97.7 | 1.024e-02
>
> Furthermore, we report results for MAS on GSO in the table below after performing a grid-search over different values for $\gamma$. We find that for $\gamma=1.05 \pm 0.13$, PowerRemap remedies Meltdown in MAS for GSO in 84.6% on average across different categories.
>
> Category | Shapes | Meltdowns | PR rescues [%]
> --- | --- | --- | --- |
> Consumer goods | 60 | 100.0 | 90.0
> Bottles, cans & cups | 23 | 100.0 | 95.7
> Unknown | 18 | 100.0 | 83.3
> Other | 29 | 100.0 | 65.5
> Total | 130 | 100.0 | 84.6
>
> Moreover, we also provide first results for MAS on SimJEB in the table below (and Table 2 in the revised manuscript) to provide further compelling evidence on the relevance of the meltdown phenomena and our test-time remedy. Again we observe that a $\gamma=1.05 \pm 0.06$ is key to provide reliable remediation of Meltdown via PowerRemap for MAS (83% remediation rate on average across all shapes).
>
> Category | Shapes | Meltdowns | PR rescues [%] | Density
> --- | --- | --- | --- | --- |
> Arch | 2 | 2 | 50.0 | 1.920e-02
> Beam | 4 | 4 | 75.0 | 1.856e-02
> Block | 9 | 9 | 100.0 | 1.371e-02
> Butterfly | 3 | 3 | 66.7 | 1.491e-02
> Flat | 11 | 11 | 90.9 | 1.370e-02
> Other | 1 | 1 | 0.0 | 8.706e-03
> Total | 30 | 30 | 83.3 | 1.488e-02
>
> From these results we conclude that the effectiveness of PowerRemap generalizes beyond GSO for both WALA and MAS. In particular, we found that the parameter $\gamma$ used in PowerRemap mostly depends on the model, as the median value for $\gamma$ varies only slightly for different shapes.
>
> ### Density
>
> To complement our results with a focus on point cloud density, Appendix E now provides an analysis of meltdown prevalence as a function of areal density. Figure 11 shows that meltdown is particularly frequent in regimes of low areal density. We are thankful to the reviewer for the valuable suggestion to include this analysis.
>
> ## Relationship between connectivity and spectral entropy (W2)
>
> We agree that spectral entropy constitutes merely correlate (proxy) of the meltdown phenomen. In Appendix H, we provide further evidence that the patterns reported in Figure 2 and 3 transfer to diverse geometries and random seeds. Additionally, we add Figure 15 which shows that the average behavior over a population of 150 diffusion seeds is consistent with the observations reported in Section 3.3. We posit that the discovery of this robust pattern is valuable to the wider interpretability/safety community, while we agree that more work is needed to understand the exact causal relationship.
>
> ## Limited experimental setup (W3)
>
> We agree with the reviewer that it is particularly interesting to investigate reconstruciton of entire scenes or measurement noise. To add more depth in this regard to our experiments, we asses whether the meltdown phenomenon and the effectiveness of PowerRemap extends beyond single-object inputs in Appendix I.1. Figure 18 shows that the failure phenomenon still occurs in this multi-object setting, and using PowerRemap reliably supresses the failure.
>
> Moving our experimental setup to reconstruction of entire scenes requires rethinking based on what measure to identify Meltdown, as the number of connected components can vary. We believe this is a fruitful direction for future research.

---

> > ### Author Response · Authors · 2025-11-22
> >
> > ## References
> >
> > [1] Laura Downs, Anthony Francis, Nate Koenig, Brandon Kinman, Ryan Hickman, Krista Reymann,Thomas B. McHugh, and Vincent Vanhoucke. Google scanned objects: A high-quality dataset of 3D scanned household items, 2022. URL: https://arxiv.org/abs/2204.11918.
> >
> > [2] E. Whalen, A. Beyene, and C. Mueller. SimJEB: Simulated Jet Engine Bracket Dataset. Computer Graphics Forum, 40(5): 9–17, August 2021. ISSN 1467-8659. doi:10.1111/cgf.14353.

---

### Official Review · Reviewer_Nujf · 2025-10-30

**Soundness:** 3
**Presentation:** 3
**Contribution:** 3
**Rating:** 6
**Confidence:** 1

**Summary:**

The paper identifies a failure mode called “meltdown” in point-cloud-conditioned 3D diffusion transformers, such as WALA and MAKE-A-SHAPE. Tiny on-surface input perturbations cause fragmented, multi-component outputs. Using activation patching, the authors localize causality to a single early cross-attention. The singular-value spectral entropy of this write tracks failure/rescue. The authors also propose PowerRemap, a test-time SVD power transform that lowers spectral entropy and substantially stabilizes outputs.

**Strengths:**

Clean activation-patching grid over depth×time pinpoints a single early cross-attention write controlling meltdown; procedure and repair map are explicit.

PowerRemap is model-agnostic, test-time only, and provably reduces spectral entropy without changing singular vectors.

On GSO, meltdown occurs widely, and PowerRemap rescues 98.3% of WALA failures.

**Weaknesses:**

For make-a-shape, reported rescue is only 10.1% with the same 𝛾, suggesting sensitivity to architecture and hyperparameters and limiting generality.

Spectral entropy is the only diagnostic evaluated; no comparison to effective rank, top-k energy, condition number, per-head concentration, or Jacobian norms.

“Connected components” may conflate legitimate multi-part objects with failures; precision/recall vs. human labels not reported.

𝛾 selection is ad-hoc (global 𝛾=100); the paper itself notes the open question of choosing 𝛾 and the speculative nature of the “consensus via low entropy” explanation.

**Questions:**

Compare spectral entropy to alternative spectral metrics for predicting meltdown

Provide an adaptive 𝛾 rule and show it fixes MAKE-A-SHAPE’s low rescue rate

---

> ### Author Response · Authors · 2025-11-22
>
> We thank the reviewer for their positive review and for the helpful, concrete suggestions to improve our manuscript. We provide detailed responses to the comments below and hoping they adress the suggestions.
>
> ## Weak results for Make-A-Shape (W1)
>
> We agree that our experimental results in its current form are limited. To provide evidence that the meltdown phenomenon generalizes beyond Google Scanned Objects (GSO) [1], we repeat our experiments on another dataset, namely SimJEB [2]. SimJEB is a curated benchmark consisting of 381 3D jet-engine bracket CAD models, and it was not included in the training data for either WALA or MAKE-A-SHAPE (MAS). We observe meltdown on SimJEB in 352 out of 381 shapes (92.4%) for WALA (see Table 1) and in 363 out of 381 shapes (95.3%) for MAS (see Table 7). We therefore conclude that the identified meltdown phenomenon exists across diverse shapes beyond GSO.
>
> We report results for WALA on SimJEB in the table below and also added it to Table 1 in our revised version. In addition to success rates we include measurements of areal density for each object category.
>
> Category | Shapes | Meltdowns | PR Rescues [%]
> --- | --- | --- | ---
> Arch | 37 | 33 | 100.0
> Beam | 46 | 46 | 100.0
> Block | 99 | 87 | 97.7
> Butterfly | 43 | 40 | 95.0
> Flat | 147 | 137 | 97.8
> Other | 9 | 9 | 88.9
> Total | 381 | 352 | 97.7
>
> Furthermore, we report results for MAS on GSO in the table below after performing a grid-search over different values for $\gamma$. We find that for $\gamma=1.05 \pm 0.13$, PowerRemap remedies Meltdown in MAS for GSO in 84.6% on average across different categories.
>
> Category | Shapes | Meltdowns | PR rescues [%]
> --- | --- | --- | --- |
> Consumer goods | 60 | 100.0 | 90.0
> Bottles, cans & cups | 23 | 100.0 | 95.7
> Unknown | 18 | 100.0 | 83.3
> Other | 29 | 100.0 | 65.5
> Total | 130 | 100.0 | 84.6
>
> Moreover, we also provide first results for MAS on SimJEB in the table below (and Table 2 in the revised manuscript) to provide further compelling evidence on the relevance of the meltdown phenomena and our test-time remedy. Again we observe that a $\gamma=1.05 \pm 0.06$ is key to provide reliable remediation of Meltdown via PowerRemap for MAS (83% remediation rate on average across all shapes).
>
> Category | Shapes | Meltdowns | PR rescues [%]
> --- | --- | --- | --- |
> Arch | 2 | 2 | 50.0
> Beam | 4 | 4 | 75.0
> Block | 9 | 9 | 100.0
> Butterfly | 3 | 3 | 66.7
> Flat | 11 | 11 | 90.9
> Other | 1 | 1 | 0.0
> Total | 30 | 30 | 83.3
>
> From these results we conclude that the effectiveness of PowerRemap generalizes beyond GSO for both WALA and MAS. In particular, we found that the parameter $\gamma$ used in PowerRemap mostly depends on the model, as the median value for $\gamma$ varies only slightly for different shapes.
>
> ## Additional spectral metrics (W2)
>
> As per reviewers suggestion, we analyze additional spectral metrics in Appendix H to assess their suitability as indicators of meltdown. Specifically, Figure 16 reports the effective rank and Figure 17 the condition number as alternatives to spectral entropy, evaluated over a diverse set of shapes. Since effective rank is a monotonic transformation of spectral entropy, we find that it performs comparably and can be used as an equally reliable indicator of meltdown. In contrast, the condition number exhibits no observable correlation with the failure phenomenon, and we therefore deem it unsuitable as an indicator.
>
> ## Limitation to single-part objects (W3)
>
> Thank you for raising this points, we clarify our definition of connected components below.
> “Connected components” refers to a purely topological notion: given the output mesh $C(G(P))$, we build its adjacency graph (triangles are neighbors if they share an edge) and define
> $$
> C(G(P)) = \text{the number of graph-connected components, computed via}\,\, \texttt{trimesh.split}.
> $$
> A “multi-part” object that is physically continuous in the mesh (e.g., a shoe with sole + upper, or a bracket with several ribs connected by faces) is therefore a single connected component ($C = 1$), even though it may have multiple semantic parts.
> In Appendix I.1 we discuss  scenarios, where input point clouds represent multiple (disconnected) objects.
>
> ## Adaptive $\gamma$-strategy (W4)
>
> Thank you for raising this valuable question.
> Indeed we observed in our newly added experiments that the poor performance of MAS on GSO was due to a suboptimal selection of $\gamma=100$. We now performed a grid-search over different values for $\gamma$ for MAS for a subset of objects of the GSO and SimJEB datasets and observed an optimal value of $\gamma=1.05 \pm 0.13$ and $\gamma=1.05 \pm 0.06$, respectively. This finding indicates that the choice of the optimal $\gamma$ is mostly dependent on the model as the variance in $\gamma$ is rather small for different shapes. In the future we envision to find a more sophisticated manner to infer the optimal value of $\gamma$ for different models.
>
> ## Questions
>
> 1. See W2
> 2. See W4

---

> > ### Author Response · Authors · 2025-11-22
> >
> > ## References
> >
> > [1] Laura Downs, Anthony Francis, Nate Koenig, Brandon Kinman, Ryan Hickman, Krista Reymann,Thomas B. McHugh, and Vincent Vanhoucke. Google scanned objects: A high-quality dataset of 3D scanned household items, 2022. URL: https://arxiv.org/abs/2204.11918.
> >
> > [2] E. Whalen, A. Beyene, and C. Mueller. SimJEB: Simulated Jet Engine Bracket Dataset. Computer Graphics Forum, 40(5): 9–17, August 2021. ISSN 1467-8659. doi:10.1111/cgf.14353.

---

> ### Comment · Reviewer_Nujf · 2025-11-28
>
> Thanks for the response. I am not entirely familiar with this area. I would like to keep positive and let the decisions be made by other expert reviewers and the AC.

---

### Official Review · Reviewer_PbKr · 2025-11-01

**Soundness:** 1
**Presentation:** 2
**Contribution:** 2
**Rating:** 2
**Confidence:** 4

**Summary:**

This paper uses mechanistic interpretability to investigates the meltdown phenomenon of 3D diffusion transformers on surface reconstruction tasks, where small perturbations to the input point cloud can cause catastrophic fragmentation of the generated 3D surfaces. Using activation patching, this paper identifies a specific cross-attention head in the WALA model whose activations have causal connections with the connectivity of the reconstructed surface. The paper shows that intervening on the magnitude of the singular values of the decomposed cross-attention output can better recover the shape of generated object. Finally, the paper connected the meltdown phenomenon with bifurcation dynamics in the reverse diffusion process.

**Strengths:**

1. The paper presents an interesting application of existing activation patching method to identify geometry-related representations within 3D latent diffusion models.
2. The proposed meltdown phenomenon is novel and well-characterized, although it remains unclear whether similar behavior would be observed on other surface reconstruction datasets beyond Google Scanned Objects (GSO).
3. The proposed PowerRemap intervention is simple yet effective, demonstrating strong recovery performance on WaLa model and the GSO dataset by intervening the cross-attention head outputs that have causal connections with output surface connectivity. Nonethelss, it is questionable whether this intervention method generalizes to other models (see weakness 2).

**Weaknesses:**

1. The generalizability of this finding is very limited. The experiment focused on two models (WaLa and MAKE-A-SHAPE) and evaluated the meltdown on only one dataset (Google Scanned Objects). It is unknown whether the meltdown phenonomon is unique to the GSO datasets, and if the cross-attention head that controls the meltdown can be found in latent 3D diffusion transformer, other than WaLa and MAKE-A-SHAPE.
2.  As shown in Tables 2 and 3 in Appendix B.3 (p. 21), the effectiveness of PowerRemap differs significantly when applied to the WALA model versus the MAKE-A-SHAPE model. While PowerRemap recovers 98% of the meltdowned generation for WALA on GSO dataset, it recovers only about 10% of meltdowned cases for MAKE-A-SHAPE model. This large discrepancy again raises concerns about the generalizability of the proposed intervention and the causal role of the identified cross-attention head in MAKE-A-SHAPE model.
3. The interpretation offered in this paper is also limited in depth. What exact geometric features have been learned by this cross-attention head? If one ablates this cross-attention head, will the meltdown phenomenon disappear? What is the trade-off between suppressing the spectral entropy of this head's output versus ablating it.
4. The trends shown in Figure 3 differ between individual and population levels. The difference at population level (across seeds) was unexplained. Within the same random seed, the plot shows that the connectivity $C$ sharply increases after the spectral entropy exceeds a threshold. However, across seeds, the meltdowns (sudden jump in $C$) occur earlier even when the spectral entropy is lower. Current text does not explain thos trend at the population level.
5. The influence of PowerRemap strength $\gamma$ on reconstruction connectivity is not studied. It remains unclear how $\gamma$ should be selected in practice or whether larger / smaller values introduce any trade-offs in reconstruction quality besides connectivity.
6. It is also unclear what data and how many data points and random seeds are used to localized the meltdown circuit in section 3.2.
7. Why search only the cross-attention outputs? The decision to restrict the search space to cross-attention outputs is insufficiently justified. Since the cross-attention outputs will be written back to the residual stream, will you obtain similar results if patch residual stream activations? How much worse does the activation patching on MLP layer outputs compared to the cross-attention outputs.

**Questions:**

1. What are the patching results for other components (MLP, self-attention, and residual stream) in the latent diffusion transformer?
2. As mentioned in Weakness 5, it is unclear what data and how many samples were used to produce Figures 2 and 3. Do the observed patterns in these plots generalize when evaluated on more data points and random seeds?
3. In Figure 3, the change in connectivity is sudden, suggesting meltdown is relatively binary phenomenon. However, the spectral entropy of the cross-attention head outputs varies smoothly. Does this imply that later MLP blocks might also contribute to (or mitigate) the meltdown failures?
4. For the qualitative examples shown in Figure 4, what are the corresponding ground-truth 3D shapes of these four objects?
5. According to Appendix B.3 (p. 20, l. 1060), the PowerRemap intervention was applied only to failure cases of the model’s generation. What happens if PowerRemap is applied to successful (non-meltdown) cases? Does it alter the output quality or connectivity in any noticeable way?

---

> ### Author Response · Authors · 2025-11-22
>
> We thank the reviewer for their thoughtful review and the constructive feedback for improving our work! We provide detailed answers to the comments below.
>
> ## Generalizability and effectiveness of PowerRemap (W1 & W2)
>
> To provide evidence that the meltdown phenomenon generalizes beyond Google Scanned Objects (GSO) [1], we repeat our experiments on another dataset, namely SimJEB [2]. SimJEB is a curated benchmark consisting of 381 3D jet-engine bracket CAD models, and it was not included in the training data for either WALA or MAKE-A-SHAPE (MAS). We observe meltdown on SimJEB in 352 out of 381 shapes (92.4%) for WALA (see Table 1) and in 363 out of 381 shapes (95.3%) for MAS (see Table 7). We therefore conclude that the identified meltdown phenomenon exists across diverse shapes beyond GSO.
>
> We report results for WALA on SimJEB in the table below and also added it to Table 1 in our revised version.
>
> Category | Shapes | Meltdowns | PR Rescues [%]
> --- | --- | --- | --- |
> Arch | 37 | 33 | 100.0
> Beam | 46 | 46 | 100.0
> Block | 99 | 87 | 97.7
> Butterfly | 43 | 40 | 95.0
> Flat | 147 | 137 | 97.8
> Other | 9 | 9 | 88.9
> Total | 381 | 352 | 97.7
>
> Furthermore, we report results for MAS on GSO in the table below after performing a grid-search over different values for $\gamma$. We find that for $\gamma=1.05 \pm 0.13$, PowerRemap remedies Meltdown in MAS for GSO in 84.6% on average across different categories.
>
> Category | Shapes | Meltdowns | PR rescues [%]
> --- | --- | --- | --- |
> Consumer goods | 60 | 100.0 | 90.0
> Bottles, cans & cups | 23 | 100.0 | 95.7
> Unknown | 18 | 100.0 | 83.3
> Other | 29 | 100.0 | 65.5
> Total | 130 | 100.0 | 84.6
>
> Moreover, we also provide first results for MAS on SimJEB in the table below (and Table 2 in the revised manuscript) to provide further compelling evidence on the relevance of the meltdown phenomena and our test-time remedy. Again we observe that a $\gamma=1.05 \pm 0.06$ is key to provide reliable remediation of Meltdown via PowerRemap for MAS (83% remediation rate on average across all shapes).
>
> Category | Shapes | Meltdowns | PR rescues [%]
> --- | --- | --- | --- |
> Arch | 2 | 2 | 50.0
> Beam | 4 | 4 | 75.0
> Block | 9 | 9 | 100.0
> Butterfly | 3 | 3 | 66.7
> Flat | 11 | 11 | 90.9
> Other | 1 | 1 | 0.0
> Total | 30 | 30 | 83.3
>
> From these results we conclude that the effectiveness of PowerRemap generalizes beyond GSO for both WALA and MAS. In particular, we found that the parameter $\gamma$ used in PowerRemap mostly depends on the model, as the median value for $\gamma$ varies only slightly for different shapes.
>
> ###  Models other than WaLa and MAKE-A-SHAPE.
>
> To the best of our knowledge, WALA and MAS are the two most recent and prevalent 3D shape generative models with diffusion transformer architectures and point-cloud conditioning. We therefore focused our analyses on these. We see the application to a wider range of 3D diffusion models as a direction for future work.
>
> ## Interpretation (W3)
>
> ### What exact geometric features have been learned by this cross-attention head?
>
> This is not a straight-forward question to answer. We are not aware of a reliable method to infer exact geometric features that have been learned by certain activations in complex Transformer architectures. The premise of mechanistic interpretability is to unveil causal relationships between model behavior and internal circuits, which is what we focused on in this work.
>
> ### Ablating cross-attention head
> We did not run any experiments for ablating the cross-attention activations entirely as it would require changes to the forward pass of the different models. However, in Figure 12 we provide results for patching the residual stream and which can indeed also remedy Meltdown. However, the effect of cross-attention on connected components is much more localized, hence it provides a fixed point of intervention that generalizes across models and datasets as we show in our revised version.

---

> ### Author Response · Authors · 2025-11-22
>
> ## Trend between individual and population level (W4)
>
> Section 3.2 shows the connectivity and spectral entropy for a simple sphere over three seeds. We now extend this analysis to four additional shapes of GSO, and two more shapes of SimJEB for WALA in Appendix G. These results are for three seeds and in Figure 15 we also provide a result for 150 seeds for a selected shape of SimJEB. We agree with the reviewer that there is variance across seeds and shapes in the jump of connected components. It is important to note that PowerRemap is invariant to the observed phase transition and results in a fully connected shape regardless of whether Meltdown occurs or not.
>
> ## Influence of $\gamma$ on $C$ (W5)
> We empirically investigate the influence of the {PowerRemap strength $\gamma$ on reconstruction connectivity in Appendix I.3. We find that for WALA PowerRemap yields  stable success rates for $\gamma > 2$ (Figure 19). However, we find that MAS is more sensitive as certain values ($\gamma = 100$, cf. Table 4) lead to suboptimal results.
>
> Overall, we find that the optimal $\gamma$ is model-dependent and independent of the data. Therefore, we recommend to practicioners to perform a swift hyperparameter-search over a small subset of shapes to find the model-dependent $\gamma$.
>
> ## Data points and random seeds to localize meltdown (Figures 2 & 3) (W6)
>
> Section 3.2 uses a sphere as an illustrative example for localizing Meltdown. We used 100 random seeds in this example. In Appendix G, we are glad to provide further evidence that the patterns reported in Figure 2 and 3 transfer to more data points and random seeds. We add Figure 13 and Figure 14 which show that the behavior transfers to diverse shapes from the GSO and SimJEB corpora. Additionally, we add Figure 15 which shows that the average behavior over a population of 150 diffusion seeds is consistent with the observations reported in Section 3.3.
>
> ## Patching different components (W7)
>
> We thank the reviewer for pointing this out. The decision to restrict the search-space to cross-attention activations is motivated by prior work [3,4,5]. To further complement that the cross-attention activation indeed exhibits a causal relationship to the Meltdown phenomenon, we conduct activation-patching for additional components (MLP, self-attention, and residual stream) of WALA in Figure 12 in Appendix G.
>
> We found no clear correlation between the Meltdown and outputs of the MLP, self-attention, and residual stream.  This corroborates the importance of cross-attention activations investigated prior work [3,4,5]. These results let us conclude that the identified cross-attention is the mechanistic source of the meltdown and also addresses the question on whether later MLP blocks might contribute to Meltdown.
>
> ## Questions
>
> 1. See W7
> 2. See W6
>
> 3. __Binary connectivity vs continuous entropy imply MLP blocks contribute to?__
>
> We don't believe that such a connection can be established based on the observed evidence. Furthermore, we now provide evidence in Figure 12d showing that patching the downstream MLP in the cross-attention block does not provide a clear correlation to the number of connected components and hence the Meltdown behavior.
>
> 4. __Ground-truth shapes in Figure 4__
>
> We have augmented Figure 4 with the respective ground-truth shapes and are thankful for this useful suggestion. The shapes resulting from the PowerRemap are semantically plausible results for the sparse point-clouds sampled from the ground truths.
>
> 5. __PowerRemap applied to successful cases__
>
> In Table 10 in the revised version, we apply PowerRemap to a selected shape of the SimJEB dataset that is not affected by Meltdown over a range of $\gamma \in \{ 2,5,10,100 \}$. We observe that Meltdown never occurs for 10 random seeds, and that the application of PowerRemap does not influence the number of connected components.
>
> We also note that, as shown in Figure 5b, PowerRemap steers the generation process to an attractor that is close-to, but not exactly the same as for non-Meltdown runs, which may produce slight shape variations. Importantly, both shapes are semantically plausible for the sparse point-clouds sampled from the ground truths (now shown in Figure 4).

---

> > ### Author Response · Authors · 2025-11-22
> >
> > ## References
> >
> > [1] Laura Downs, Anthony Francis, Nate Koenig, Brandon Kinman, Ryan Hickman, Krista Reymann,Thomas B. McHugh, and Vincent Vanhoucke. Google scanned objects: A high-quality dataset of 3D scanned household items, 2022. URL: https://arxiv.org/abs/2204.11918.
> >
> > [2] E. Whalen, A. Beyene, and C. Mueller. SimJEB: Simulated Jet Engine Bracket Dataset. Computer Graphics Forum, 40(5): 9–17, August 2021. ISSN 1467-8659. doi:10.1111/cgf.14353.
> >
> > [3] Surkov, V., Wendler, C., Mari, A., Terekhov, M., Deschenaux, J., West, R., Gulcehre, C., & Bau, D. (2025). One-step is enough: Sparse autoencoders for text-to-image diffusion models. arXiv:2410.22366. https://arxiv.org/abs/2410.22366
> >
> > [4] Tang, R., Liu, L., Pandey, A., Jiang, Z., Yang, G., Kumar, K., Stenetorp, P., Lin, J., & Ture, F. (2022). What the DAAM: Interpreting Stable Diffusion using cross-attention. arXiv:2210.04885. https://arxiv.org/abs/2210.04885
> >
> > [5] Tinaz, B., Fabian, Z., & Soltanolkotabi, M. (2025). Emergence and evolution of interpretable concepts in diffusion models. arXiv:2504.15473. https://arxiv.org/abs/2504.15473

---

### Author Response · Authors · 2025-11-22
**Global Response to Review Comments**

We thank the reviewers for their valuable feedback that helps us improve our work. We are glad that the reviewers highlighted our work relevant (PbKr, Nujf) and fascinating (5ZNt), our insights novel and interesting (PbKr, 9Hbz, 5ZNt), our method simple and efficient (Nujf), our experimental evidence strong (Nujf), and the connection to diffusion dynamics interesting (9Hbz, 5ZNt).

We found that there is generally a consensus in the major  concerns raised by the different reviewers, which we address as follows:
- __Effectiveness__: We improved the performance of PowerRemap for Make-A-Shape on GSO by searching over different values for $\gamma$, significantly improving test-time remediation rate from 10\% to 84.6\% on average.
- __Generalizability__: We added results for WALA and Make-A-Shape on SimJEB. PowerRemap consistently remediates Meltdown in 97.7\% and 83.3\% for WALA and Make-A-Shape, respectively. This highlights generalizability across datasets.
- __Clarifications__ with respect to Meltdown localization on seed vs population level and its appearance with varying density levels. Investigations on more seeds, more shapes, and varying densities corroborate our originally reported findings.

We further addressed all remaining questions in the individual responses.

---

### Author Response · Authors · 2025-12-03
**On the OpenReview situation**

Dear AC, SAC, PC,

In light of the OpenReview incident, we fully support the Program Chairs’ decision to uphold the integrity of the review process. We know this adds extra work for newly appointed Area Chairs and are very grateful for your time and effort.

Before the discussion was frozen, we submitted a detailed rebuttal addressing all concerns raised to the best of our knowledge. Unfortunately, there was no further engagement with our responses. We would kindly ask you to consider our **detailed rebuttals**, which directly address all issues raised in the initial reviews as well as the **revised manuscript**, which includes clarifications and extra experiments (with changes highlighted in blue for convenience). Our updates are summarized as follows:

- **Generality**: Added results for WALA and MAKE-A-SHAPE on the SimJEB dataset, where PowerRemap remediates meltdown in 97.7% and 83.3% of cases, respectively, demonstrating that both the phenomenon and our remedy generalize across datasets and models.

- **Effectiveness**: Improved the performance of PowerRemap for MAKE-A-SHAPE on GSO by searching over different values of $\gamma$, increasing the test-time remediation rate from 10% to 84.6% on average.

- **Clarifications**: Provided clarifications with respect to meltdown localization on seed vs population level, and studied its behavior across varying density regimes. Experiments with more seeds, more shapes, and different densities corroborate our originally reported findings.

Despite the circumstances, we believe the process has significantly strengthened our submission. Thank you again for stepping in under these unusual conditions.

Best,
The Authors

---

### Meta-Review · Area_Chair_sTMa · 2026-01-07

**Summary:**

This paper investigates two state-of-the-art diffusion transformers for 3D surface reconstruction, WALA and MAKE-A-SHAPE on the task of surface reconstruction from sparse point clouds, and observe an intriguing failure mode. Then the authors employ activation patching to test the causal role of activations with respect to meltdown and provide interpretation. The reviewers agree the proposed meltdown phenomenon is novel, but raised concerns about the insufficient experiments, limited generalizability of the finding, and lack of clarification of experimental settings.

The authors addressed reviewers' initial concerns of the work during the rebuttal and discussion period, but the technical contents can be further improved.

**Reviewer Concerns:**

Part of concerns were addressed and discussed by the rebuttal.

**Reviewer Scores:**

I think reviewers may keep their initial scores.

---

### Decision · Program_Chairs · 2026-01-26

Reject